# Atmospheric response to cold wintertime Tibetan Plateau conditions over East Asia in climate models

Alice Portal[1,2,3], Fabio D'Andrea[3], Paolo Davini[4], Mostafa E. Hamouda[5,6], and Claudia Pasquero[2,4]

[1]Institute of Geography, Oeschger Centre for Climate Change Research, University of Bern, Bern
[2]Department of Earth and Environmental Sciences, Università di Milano - Bicocca, Milan, Italy
[3]Laboratoire de Météorologie Dynamique/IPSL, École Normale Supérieure, PSL Research University, Sorbonne Université, École Polytechnique, IP Paris, CNRS, Paris, France
[4]Consiglio Nazionale delle Ricerche, Istituto di Scienze dell'Atmosfera e del Clima (CNR-ISAC), Torino, Italy
[5]Astronomy and Meteorology Department, Faculty of Science, Cairo University, Cairo, Egypt
[6]Institute for Atmospheric and Environmental Sciences, Goethe University Frankfurt, Frankfurt am Main, Germany

**Correspondence:** Alice Portal (a.portal@campus.unimib.it)

**Abstract.** Central Asia orography (namely the Tibetan and Mongolian plateaux) sets important features of the winter climate over East Asia and the Pacific. By deflecting the mid-latitude jet polewards it contributes to the formation of the Siberian High and, on the lee side, to the advection of dry cold continental air over the East Asian coast and the Pacific Ocean, where atmospheric instability and cyclogenesis thrive. While the mechanic forcing by the orography is assessed in a number of modelling studies, it is still not clear how near-surface temperature over the two most prominent orographic barriers of the Central Asian continent - the Tibetan and Mongolian plateaux - influences the winter climate. The problem is particularly relevant in view of a well known cold bias in state-of-the art climate models in proximity of the Tibetan Plateau, likely related with the modelling of land processes and land–atmosphere interaction over complex orography. Here we take advantage of the large spread in near-surface temperature over the Central Asia plateaux within the Coupled Model Inter-comparison Project, Phase 6 (CMIP6) to study how colder-than-average Asian plateaux temperatures impact the atmospheric circulation. Based on composites of the CMIP6 models' climatologies showing coldest Tibetan Plateau conditions, we find that such negative temperature anomalies appear to amplify the atmospheric response to orography, with an intensification of the East Asian winter monsoon and of the equatorward flank of the Pacific jet. The results of the CMIP6 composite analysis are supported by experiments run with an intermediate-complexity atmospheric model, forced by a similar pattern of cold surface temperatures over the Central Asia plateaux. Within this setting, the relative influence of the Tibetan and the Mongolian Plateau surface conditions is analysed. Based on the results reported in this work we prospect that advances in the modelling of the land energy budget over the elevated regions of Central Asia could improve the simulation of the East Asia / Pacific sector climate, together with the reliability of climate projections and the performance of shorter term forecasts.

**Short non-technical summary.** The differences between climate models can be exploited to infer how specific aspects of the climate influence the whole Earth system. This work analyses the effects of a negative temperature anomaly over the Tibetan Plateau and its surroundings on the winter atmospheric circulation. We show that models with a colder-than-average

Tibetan Plateau present a reinforcement of the East Asian winter monsoon and we discuss the atmospheric response to the enhanced transport of cold air from the continent toward the Pacific Ocean.

# 1   Introduction

The impact of orography on the extratropical circulation was proposed by the analytical studies of Charney and Eliassen (1949) and Bolin (1950), while Smagorinsky (1953) first discussed the matching of orographic and thermal forcing by land-sea contrast in order to explain the longitudinal variations of the mid-latitude westerlies. Manabe and Terpstra (1974) and Hahn and Manabe (1975) analysed the impact of the Tibetan Plateau on the Asian climate by running an atmospheric general circulation model (AGCM) with and without mountains. They proved that the elevation of Central Asia is essential to reproduce the position and strength of the low-level winter anticyclone known as the Siberian High and for the maintenance of the South-East Asian summer monsoon, which, thanks to the intense uplift from orography, extends from the Indian sector as far as East Asia. The regional dryness and humidity of the aforementioned winter and summer circulation patterns and their association with orography were examined by Broccoli and Manabe (1992).

More recently, starting with Sato (2009), the influence of lower elevation mountain chains on the Asia and Pacific climate has been considered separately from that of the Tibetan Plateau. This applies in particular to the mountain chains extending north east of Tibet. Similarly to White et al. (2017), we denote the orography between approximately 20 to 40° N and 62 to 120° E as the Tibetan Plateau or TP region (green box in Figure 2), and that between approximately 38 to 60° N and 65 to 140° E as the Mongolian Plateau or MP region (orange box in Figure 2).

In the cold season the East Asia / Pacific circulation is dominated by the East Asian winter monsoon, which consists in north-westerly advection of cold dry continental air from Siberia off the Asian coast (Zhang et al., 1997; Chan and Li, 2004). The winter thermal emission of the TP land and of the air column above generate a tropospheric heat sink over the Plateau (Yanai et al., 1992; Yanai and Wu, 2006; Duan and Wu, 2008) that reinforces the Eurasian mid-tropospheric thermal high (Shi et al., 2015). Moreover, the presence of TP and MP orography reduces the westerlies upstream and enhances the north-westerly winds over East Asia and the Pacific (Shi et al., 2015; Sha et al., 2015). On the lee side of the plateaux, the cold continental advection modulates the thermal contrast with the Pacific Ocean and the local baroclinicity, which fuel the Pacific jet stream downstream (Shi et al., 2015; White et al., 2017). Notwithstanding the lower elevation and extension of the MP compared to the TP, the MP is more relevant for the winter circulation because of its ideal position - in terms of impinging low-level winds and meridional potential vorticity gradients - for acting as a source of Rossby waves (Held and Ting, 1990; White et al., 2017).

Conversely, the warm season circulation is driven by the East Asian summer monsoon, modulating rainfall over land and ocean (Yihui and Chan, 2005). This is sustained in strength and extension by the atmospheric uplift produced by Asian orography, which constitutes a tropospheric heat source in summer (Yanai et al., 1992; Hahn and Manabe, 1975; Ye and Wu, 1998). The orographic control over the summer monsoon is mostly accomplished by the TP - the MP playing only a marginal role; this, among other things, reinforces the monsoonal circulation and the associated precipitation along the east coast of Asia (see Figures 6, 9, 10 in Sha et al., 2015).

Considering the importance of the Central Asian orography for the climate of the Asia / Pacific sector, it is not surprising to find examples in literature where orographic surface and near-surface thermal conditions (acting as tropospheric heat sources or sinks, Yanai et al., 1992) have an impact on the circulation downstream. Indeed, evidence is found for the relevance of spring and summer temperatures over Asian orography for the atmospheric conditions far downstream (see Wu et al. (2015) for a review and Xue et al. (2021, 2022) for recent work on the impact of spring TP land initialisation in subseasonal-to-seasonal predictions). In the extended winter season (October–March) the presence of anomalous snow cover changes the tropospheric energy budget through an increase of the surface albedo, enhancing the reflection of shortwave radiation and the cooling of the land surface and the atmosphere (Yeh et al., 1983). Analyses on the dynamical influence of Tibetan Plateau snow cover indicate that it is relevant for the atmospheric circulation at intraseasonal time scales (Li et al., 2018) and that, when anomalies are persistent, it may modulate interannual variability (Chen et al., 2021; Clark and Serreze, 2000; Henderson et al., 2013) and long-term projections (Liu et al., 2021). In a more idealised context, winter positive thermal forcing over mid-latitude land - as in a climate with a reduced winter land-sea thermal contrast caused by the faster warming of continents with respect to oceans - was analysed by Portal et al. (2022). It was shown there that the atmospheric response to idealised warming over East Asia (including the orography) dominated over a pattern of similar intensity imposed over the North American continent. The work by Henderson et al. (2013), comparing snow-induced temperature forcing over the two continents, reaches similar conclusions regarding the relevance of East Asian surface conditions for the Pacific sector. A possible explanation for this is that the elevated Asian forcing, releasing heat directly in the mid troposphere, is more effective in producing a large hemispheric response than an equivalent lower-level forcing over North America (Hoskins and Karoly, 1981; Trenberth, 1983; Ting, 1991). Notwithstanding the potentially high impact of anomalous surface conditions over the Tibetan and Mongolian plateaux on the East Asian climate, their dynamical role has been poorly investigated.

An additional motivation to approach the topic of thermal forcing over the Asian plateaux is the presence of a significant multi-model mean (MMM) temperature bias in the region of East Asia, which is evident over successive phases of the CMIP and over multiple seasons. Priestley et al. (2022) detect a strong deviation from the reanalysis temperature in the summer season and, based on the modified thermal gradients in the lower troposphere, hypothesise a role of the TP land temperature on the baroclinicity and cyclogenesis downstream. Along the same lines, East Asian winter conditions are anomalously cold among several climate models (Wei et al., 2014; Gong et al., 2014), although improvements, associated with a closer representation of the winter monsoon, have been detected in the transition from CMIP Phase 3 to Phase 5 (Wei et al., 2014). The winter bias is specially strong over the TP region (Figure 1 and Peng et al., 2022; Fan et al., 2020), where limited progress was obtained in the transition from CMIP5 to CMIP6 (Lun et al., 2021; Hu et al., 2022). These studies also highlight the presence of a wide inter-model spread in year-round East Asian and TP temperatures among the CMIP climate models, which appears to be related with the difficulties in representing surface energy fluxes (Wei et al., 2014), in particular over regions characterised by complex orography and seasonal variations in snow cover (e.g. Su et al., 2013; Chen et al., 2017; Li et al., 2021).

The cold Tibetan Plateau temperature bias has been examined in some detail by Chen et al. (2017). Among the climate models taking part in CMIP5 they identify a strong bias in the western region of the Plateau (consistent with Figure 1(a)) and show that it is more evident in terms of near-surface than surface (skin) temperature. The reason for the emergence of the strong

near-surface bias is investigated by decomposing the different contributions to the low-level energy budget. Anomalous snow cover corresponds to an increase in the surface albedo, hence in the reflection of shortwave radiation, and this is anti-correlated with upward turbulent heat fluxes. While the surface temperature is weakly affected by these terms, due to compensation between incoming shortwave radiative and outgoing turbulent fluxes, a reduction in the turbulent heat flux into the atmosphere, leading to a decrease in the low-level water vapour content and thermal radiation, cools the boundary layer. By identifying physically interlinked low-level and surface processes modifying the energy budget, Chen et al. (2017) are able to explain why several CMIP5 models present a low-level cold bias over the Tibetan Plateau. These findings are likely applicable to the CMIP6 models affected by similar TP temperature biases (Lun et al., 2021; Hu et al., 2022).

In the present paper, by analysing the implications of cold Central Asia orography winter conditions for the large-scale circulation on the lee side of the mountains, the possible dynamical consequences of the recurrent Tibetan Plateau cold bias in climate models are explored. To do this we take advantage of the large temperature spread detected over TP and MP among CMIP6 models to construct a multi-model realisation of a cold anomaly. The atmospheric circulation in such "cold TP composite" is analysed in the Asia / Pacific sector, taking into account the East Asian winter monsoon. The results obtained from the multi-model study are further tested with an intermediate-complexity Atmospheric General Circulation Model (AGCM) forced by land-surface temperature patterns taken from the anomalies in the CMIP6 "cold TP composite". Finally, to isolate the individual role of the Mongolian and Tibetan plateaux in the atmospheric response to cold Central Asia orography, we consider two separate AGCM experiments where MP or TP forcing are compared against a widespread TP and MP forcing. The two approaches (CMIP6 compositing and AGCM idealised simulations) are described in the Methods, the outcomes and their mutual consistency are examined and discussed in the Results and a final summary is provided in the Conclusions.

## 2   Methods

### 2.1   CMIP6 simulations

We use CMIP6 coupled historical simulations over years 1979–2008 and we compute the January-February climatology over the 30-year period, taken as representative of the recent climate. The results for January-February - referred to as *winter* throughout the paper - are equivalent to those considering December-January-February, while, as in Clark and Serreze (2000), results for an extended October–March winter are weaker in intensity (not shown). We select one member per climate model from the CMIP6 dataset, as specified in Table 1, to obtain a sample of 37 historical simulations. Based on an index of Tibetan Plateau temperature (i.e. the climatological weighted-area average of near-surface temperature in the black box of Figure 2(b), comprising latitudes 25 to 40° N and longitudes 70 to 105° E over the period 1979–2008), the six simulations with temperature below -1.0 sdev (standard deviations) from the CMIP6 multi-model mean (MMM) form the "cold TP composite" (see models highlighted in bold in Table 1). The composite fields are shown in terms of the anomalies from the climatology of the MMM, with stippling where the anomalies exceed the 95th percentile of a random distribution, computed from 1000 samples of 6-model composites extracted randomly and without repetition from the 37 model realisations (Wilks, 2011). Stippled anomalies (as defined above) are referred to as *significant* within the text. As a caveat, it is important to note that the "cold TP composite"

is composed of multiple models from the CNRM and CCCma institutions (see Table 1). However, a similar selection method (simulations with TP temperature below a threshold of -1.0 to -0.75 sdev from the MMM), but based on a single model per institution, produces consistent results, extending to the cases when additional modelling systems are included in the composite (BCC-CSM2-MR and FIO-ESM-2-0 for the -0.75 sdev threshold selection).

Wind components and air temperature at levels between 1000 and 700 hPa and at 300 hPa are extracted from the CMIP6 archive and used in the analysis. Turbulent surface heat fluxes, surface temperature (skin temperature or SST for open ocean) and near-surface temperature (usually 2-meter air temperature) are also used. Due to the lack of availability of daily fields for a large subset of the CMIP6 models, the analyses on the "cold TP composite" are based on monthly-mean variables averaged in model climatologies. Moreover, we report that surface latent heat flux in KIOST-ESM, meridional wind and temperature advection in CAS-ESM2-0, zonal wind, temperature advection and Eady growth rate in FGOALS-f3-L are excluded from the composite analysis because of the inaccessibility of the datasets in the servers providing the CMIP6 archive.

## 2.2 Idealised experiments

To confirm the link between surface temperature and circulation anomalies in the CMIP6 compositing exercise we run idealised experiments using an 8-level AGCM developed at the International Centre for Theoretical Physics (ICTP), and known as SPEEDY for Simplified Parametrization, primitivE-Equation DYnamics. The model is spectral on the sphere, with triangular truncation at total wavenumber 30 (T30) and a Gaussian grid of 96 by 48 points, and includes simple parametrisation of moist processes (Molteni, 2003). Despite the low horizontal and vertical resolution, SPEEDY displays an adequate performance for the analysis of large-scale features of the climate system (Kucharski et al., 2006, 2013). SPEEDY is run in perpetual-winter mode (200 January months and 200 February months) with prescribed sea-surface temperatures (SSTs), sea-ice cover (SIC) and land-surface temperatures (LSTs). Two types of simulations are considered:

- a *control integration* where SST and SIC are equal to the 1979–2008 HadISST climatologies (Rayner et al., 2003). The LST corresponds to the January and February climatology from a separate SPEEDY 10-member ensemble, which is run with a freely evolving LST scheme and with prescribed climatological SIC and evolving SSTs 1979–2008 from HadISST. Details on SPEEDY's LST scheme are available in the Appendix B of Portal et al. (2022);

- three *cold integrations* with SST and SIC as in the *control*, and with LST forcing corresponding to the significant anomalies of surface temperature from the "cold TP composite" within 60–140° E and 20–60° N ("TP+MP experiment") or within 60–140° E and 38–60° N ("MP experiment") or within 60–140° E and 20–38° N ("TP experiment", smoothed by $\exp\{-\frac{1}{2} \cdot \frac{(\phi - 38° \, \text{N})}{(5° \, \text{N})^2}\}$ where the latitude $\phi$ is greater than 38° N). The patterns have been interpolated onto SPEEDY's grid (see Figure 4(a,e,i)).

The responses to "TP+MP", "MP" and "TP" forcing experiments visualised in the Results correspond to the climatological difference "*cold - control*", averaged over January and February. The stippling indicates anomalies exceeding the 95th percentile of a reference distribution obtained for each experiment by randomly permuting 1000 times a pool of daily fields composed by the "cold" and "control" integrations together. The fields are distinguished by month but not by forcing, in order to obtain 1000

samples of the average January-February "(*cold - control*)$_{\text{perm}}$" anomaly (Wilks 2011). Stippled anomalies (as defined above) are referred to as *significant* within the text.

## 2.3 Diagnostics

We introduce here the diagnostics used in the analysis of the results:

- temperature advection is

$$-\boldsymbol{u} \cdot \nabla T = -\left( u\frac{\partial T}{\partial x} + v\frac{\partial T}{\partial y} \right),$$

  where $\boldsymbol{u} = u\hat{\mathbf{i}} + v\hat{\mathbf{j}}$ is the horizontal wind composed of the zonal $u$ and meridional $v$ components and $T$ is the temperature;

- the Eady growth rate corresponds to

$$\sigma = 0.31 \, f \, \frac{du}{dZ} \, \mathcal{N}^{-1},$$

  where $f$ is the Coriolis parameter, $Z$ is the geopotential height and $\mathcal{N} \equiv \sqrt{(g/\theta)\,d\theta/dz}$ is the Brunt-Väisälä frequency with $\theta$ potential temperature and $g$ Earth's gravitational acceleration.

Both quantities are computed using mean climatological variables, giving the *temperature advection by the mean flow* and the *Eady growth rate of the mean state*.

The following diagnostics are computed on SPEEDY integrations only:

- meridional eddy momentum flux (MEMF), the product of the 2–6 day Fourier filtered wind components $u^{\text{HF}}v^{\text{HF}}$. Its meridional convergence ($-\frac{\partial}{\partial y}\left(u^{\text{HF}}v^{\text{HF}}\right)$) represents the dominant term of eddy momentum deposition in the zonal flow (Hoskins et al., 1983);

- eddy total energy flux (TEF, Drouard et al., 2015), used to estimate the downstream propagation of eddy total energy, and defined as

$$\text{TEF} \equiv \boldsymbol{u} \cdot (\text{EKE} + \text{EAPE}) + \boldsymbol{u}_a^{\text{HF}} Z^{\text{HF}}, \qquad \boldsymbol{u}_a \equiv \boldsymbol{u} - \frac{g\hat{\mathbf{k}}}{f} \times \nabla Z.$$

  The contributions come from the advective flux of $\text{EKE} \equiv [(u^{\text{HF}})^2 + (v^{\text{HF}})^2]/2$ (eddy kinetic energy) and of $\text{EAPE} \equiv (h^2/s^2)(\theta^{\text{HF}})^2/2$ (eddy available potential energy)[1], and from the ageostrophic geopotential flux. The latter is defined in terms of $Z$ (geopotential height) and $\boldsymbol{u}_a$ (ageostrophic horizontal wind).

The climatological MEMF and TEF are computed on high-pass filtered daily fields (represented with superscript $^{\text{HF}}$), averaged over the total time duration of the SPEEDY integrations.

---

[1]The EAPE parameters $s^2 = -h\,\partial\theta_{cl}/\partial p$ and $h = (R/p)(p/p_s)^{R/C_P}$ depend on pressure ($R$ is the gas constant, $p_s$ is 1000 hPa, $C_p$ is the specific heat of the air at constant pressure) and on the potential temperature climatology $\theta_{cl}$.

## 3 Results

The representation of the winter (January-February) near-surface temperature climatology by CMIP6 models in the historical period 1979–2008 shows a cold bias over the Arctic and over many inland regions of the Northern Hemisphere, including most of East Asia, with a peak in the mid-west of the Tibetan Plateau (Figure 1(a)). Panel (b) of Figure 1 shows the average bias in the TP box (black box in Figure 2(b)) for each of the CMIP6 models: apart from a few exceptions, models are colder than the reanalysis, and those belonging to the same institutions show consistent values. Otherwise, warm biases are detected in the entrance regions of the storm tracks, over north-east Siberia, in some areas of the Middle East, in the far-west (Hindu Kush) and far south-east (Hengduan Mountains) of the Tibetan Plateau.

The amplitude of the inter-model spread in near-surface temperature (computed in terms of standard deviation) is displayed in Figure 2(a). The spread is generally larger over land than over ocean and grows with latitude, with largest amplitude attained around and poleward of the 60° N latitude circle, and with a maximum over the Atlantic and Pacific Oceans which is likely due to the inter-model variability in the position of the winter sea-ice cover boundary. An additional mid-latitude hot-spot can be easily identified in the Tibetan Plateau, extending north over the Mongolian Plateau (cf. temperature spread and green and orange boxes over orography in Figure 2(a,b)). The high orographic elevation of this region implies that near-surface turbulent fluxes are released deeper in the mid-latitude atmosphere, where heat sources and sinks are known to result in stronger circulation responses (Trenberth, 1983). Motivated by this, and by results confirming the sensitivity of the winter mid-latitude circulation to East Asian surface conditions (e.g. Portal et al., 2022; Henderson et al., 2013; Cohen et al., 2001), we analyse in the following the dynamical features of a "cold TP composite". The composite corresponds to the average of a model selection based on a TP temperature index, i.e. the area-weighted spatial and temporal mean of near-surface temperature over the black box in Figure 2(b). The specified region is characterised by large temperature spread and high elevation within the Tibetan Plateau domain; the biases in the TP temperature index with respect to reanalysis are displayed in Figure 1(b).

A selection of surface variables from the "cold TP composite" is presented in Figure 3. The near-surface temperature map features an intense cold anomaly over the orography of Central Asia, peaking over the TP and extending north-eastwards to the MP. We note that significant surface anomalies are found also elsewhere in the North America / North Atlantic sector, however, since our focus is on Asian orography and its downstream impacts, regional signals that are unlikely to interact with the Asia / Pacific sector are neither presented nor discussed. By comparing the surface and near-surface temperature patterns over the TP (cf. Figure 4(a) and Figure 3(a)), we notice that the surface temperature anomaly is stronger in intensity than the near-surface anomaly, and conclude that in the "cold TP composite" land has a cooling effect on the atmosphere above. This is corroborated by negative anomalies of surface sensible and latent heat flux, which correspond to reduced latent and sensible warming of the atmosphere in regions where the MMM fluxes are weakly positive, or to enhanced atmospheric cooling by the surface where the MMM fluxes are negative (Figure 3(b,c)). The significant signal in sensible heat flux is strong over the center of the TP, while the latent heat flux term is significant elsewhere over the TP and MP regions.

In the "cold TP composite" anomalous snow amount is detected in correspondence with the strongest sensible heat flux anomalies, but is not reported because data is available only for a limited sub-group of models. Although here we cannot verify

the role of snow in the low-level energy budget, the anomalies in the surface variables are coherent with each other and with the results in Chen et al. (2017). They show that over the TP the processes causing cold biases may involve anomalous snow enhancing the surface albedo with negative effects on the low-level water vapor content and the downward longwave radiation, which ultimately result in a cooling of the boundary layer. The existence in CMIP6 models of a variety of schemes for land, snow and atmospheric boundary layer and of the mutual interaction between these over complex orography, are likely at the origin of the wide inter-model spread over the TP. In support of this view, the surface temperature anomalies do not appear to be driven by the circulation upstream of the TP (Figure 4 and 5).

The low-level temperature and wind conditions of the CMIP6 "cold TP composite" are shown in Figure 4(a–d). We note that at 850 hPa the negative thermal anomaly extends north-eastward of the most elevated area of the Tibetan Plateau - represented by grey patching - and reinforces the thermal cooling induced by the uplift of MP orography (cf. to Figure 11 in Sha et al., 2015). East of this region the westerly zonal winds of the Pacific eddy-driven jet are reinforced (Figure 4(c)). At the same time, the southward wind anomaly over East China and the northward wind anomaly over the Pacific ocean (Figure 4(d)) intensify the cyclonic circulation over the Asian coast and consequently also the East Asian winter monsoon. Typical features relatable to a strong East Asian winter monsoon are captured by the sea-level pressure and mid-tropospheric geopotential height fields in Figure 5 (cf. with strong and weak monsoon conditions in Figure 6 of Jhun and Lee (2004)). A deeper zonal pressure contrast to the east of the Siberian High (Figure 5(a)) and a lower 500 hPa isobaric surface over the Asian coast (Figure 5(b)) reinforce the 300 hPa jet over land and south of Japan (Figure 5(c)), and adhere to maps describing the atmospheric state associated with an intense monsoon.

A comparison of the results in Figure 5 with the maps in Sha et al. (2015) and Shi et al. (2015) shows that the cooling over Central Asia orography amplifies the atmospheric response to orography itself. The positive interference between orographic forcing and superposed cooling corresponds closely to the outcome of a set of idealised experiments by Ringler and Cook (1999), featuring combinations of mechanic orographic and thermal forcing under varying mean-flow conditions.

The advection of cold air downstream of the TP (Figure 6(a)) is supported by the negative temperature anomaly on the orography and, to the east, by the reinforcement of the north-westerly wind (Figure 4(b,d)). These conditions are responsible for intensified meridional temperature gradients east of the TP and along the Pacific coast which enhance the baroclinicity (see positive anomalies in the Eady growth rate west and east of the Chinese coastline at latitudes 20–40° N, Figure 6(b)). Given that the Eady growth rate measures the environmental conditions favourable to atmospheric baroclinic instability (see definition in Methods), we expect the strengthening of the jet at the entrance of the Pacific basin (Figure 4(c)) to be induced by increased eddy momentum deposition east of the TP and over the East China Sea., regions where cyclogenesis is climatologically high in mid winter (Priestley et al., 2020; Schemm et al., 2021). An analysis of the eddy feedback on the zonal flow for the idealised "TP+MP experiment" - generally coherent with the results of the CMIP6 composite analysis - supports the hypothesis that the jet strengthening is induced by an intensification (weakening) of the synoptic activity upstream and to the south (north) of the jet maximum. This will be described in more detail in the paragraphs dedicated to the idealised experiments.

The strong surface heat flux anomalies present over the Pacific basin in the "cold TP composite" (Figure 3(b,c)) are related to the strengthening of the Pacific jet over and downstream of the East China Sea (Figure 4(c)), which extend down to the near-

250 surface level (green arrows in Figure 3(c)) and intensify the advection of cold air masses over the ocean (Figure 6(a)). Indeed, cold air temperatures and strong winds in the boundary layer reinforce the surface turbulent heat fluxes by the sea surface. We note that the relation between (i) cold TP temperatures, (ii) strong low-level winds entering the Pacific basin south of Japan and (iii) strong sensible heat fluxes from the ocean surface over the South China Sea, shows a linear tendency across the CMIP6 models (e.g. the correlation coefficient between (i) and (iii) is -0.85, where (i) is the near-surface temperature in the TP box

(black box in Figure 3(b)) and (iii) is the surface sensible heat flux in a [25-40° N, 120-135° E] box). This confirms that the impact of the TP thermal conditions on the western North Pacific region is not just a peculiarity of the "cold TP composite", but rather extends to the whole CMIP6 ensemble. In the composite we also observe a significant decrease in the latent heat flux east of 135° E (Figure 3(c)) associated with a downstream weakening of the jet (outside the maps' boundaries). The origin of the negative latent heat flux anomaly may be related to a subtropical or tropical Pacific signal emerging from the selection of

CMIP6 models (Figures 3(a–c), 4(b–d)).

To argue for the existence of a causal relation linking the cold Asian orography and the enhancement of the East Asian winter monsoon we run an idealised experiment using the model SPEEDY (a perpetual winter simulation with prescribed surface temperatures, for details see Section 2). The response of SPEEDY to "TP+MP" forcing - a surface cooling over Central Asia orography (Figure 4(e)) resembling the pattern of the "cold TP composite" (Figure 4(a)) - in terms of air temperature,

zonal wind and meridional wind at 850 hPa is shown in panels (f–h) of Figure 4. As in the CMIP6 composite, we find a cold anomaly to the north-east of the TP, with enhanced north-westerly winds downstream of the mountain barrier advecting cold air onto East Asia and over the Pacific (Figure 7(a)). While in the CMIP6 composite the significant strengthening of the jet terminates at about 160 E, in the "TP+MP experiment" the strengthening is zonally coherent over the Pacific basin. The positive meridional wind signal over the North Pacific is also different, with a weak positive anomaly limited to the high latitudes in

the SPEEDY experiment (Figure 4(h)), contrasting with a strong positive anomaly extending from 20 to 70° N in the CMIP6 composite (Figure 4(d)). These discrepancies might be related to the presence of additional signals emerging from the selection of CMIP6 models, such as Pacific tropical forcing and cold North America land temperatures, or from the difference between the MMM and the SPEEDY climatology. Nonetheless, they do not undermine the striking similarity between the "cold TP composite" and the response of the "TP+MP experiment" (cf. panels (b–d) and (f–h) in Figure 4). As previously noted for the

CMIP6 composite, also the response to "TP+MP" cooling corresponds to an intensification of the East Asian winter monsoon (cf. Jhun and Lee, 2004) and to a positive interference with the atmospheric response to mountain uplift (cf. Shi et al., 2015).

In the "TP+MP experiment" the increase in the low-level baroclinicity to north-east of the TP and over the Pacific Ocean at latitudes lower than 40° N (Figure 7(b)), affects the upper-level synoptic activity. The pattern of meridional eddy momentum flux (MEMF, Figure 7(c)), which is climatologically negative to the north of the storm track and positive to its south (see e.g.

Hoskins et al., 1983), shifts equatorwards. The zonal convergence of meridional eddy momentum is also displaced to the south, and increases inland to the north-east of the TP (negative purple contours in Figure 7(c)), where it reinforces the jet across the tropospheric column (cf. green contours in Figure 8(b) and shading in Figure 4(c)). Contrarily, the wind in the northern flank of the jet, experiencing reduced momentum convergence from the synoptic disturbances, weakens. The flux of eddy total energy

(TEF, Figure 8(b)), representing the propagation of eddy energy along the storm track, confirms the increase (decrease) in the synoptic activity in correspondence of the region of jet intensification (slowdown).

In the papers by White et al. (2017) and Sha et al. (2015) the winter NH circulation is shown to be more impacted by the presence of the MP than by the TP, because of the former's latitudinal position and of its interaction with the Pacific low-level jet (Held and Ting, 1990). We briefly consider the role of thermal anomalies over the two regions by showing the results of two experiments. In the "MP experiment" the cold anomalies from the "TP+MP experiment" north of $38°$ N are selected (Figure 4(i)). In the "TP experiment" the anomalies south of $38°$ N are selected, by applying the function $exp\{-\frac{1}{2} \cdot \frac{(\phi - 38° \text{ N})^2}{(5° \text{ N})^2}\}$ where latitude $\phi$ is greater than $38°$ N (Figure 4(m)); the smoothing function, although causing some superposition of the "MP" and "TP" forcing patterns (panels (i,m) of Figure 4), is necessary to avoid numerical divergences generated by steap meridional temperature gradients.

The low-level response to "MP" forcing shows cold anomalies limited to high mid latitudes (Figure 4(j)) and cold advection centered over Japan (Figure 7(d)). The baroclinicity is enhanced at higher latitudes with respect to the "TP+MP experiment" (cf. panels (b) and (e) of Figure 7). Coherently with the changes in the meridional temperature gradients (baroclinicity), and notwithstanding a weak decline in the upper-level eddy energy over the Pacific Ocean north of $40°$ N (Figure 8(c)), MP cooling strengthens the Pacific jet around its maximum intensity (green contours in Figure 8(c) and shading in Figure 4(k)). Although the results show that thermal forcing on the MP is relevant for the climate of the Pacific sector, the position of the forcing is not appropriate to have consistency with the response of the "TP+MP experiment", hence with the anomalies emerging in the "cold TP composite".

On the other hand, the "TP experiment" shows strong similarity with the "MP+TP experiment". It features strong advection of cold temperatures to the south of Japan (Figure 7(g) and 4(n)) which produces baroclinic conditions south of $40°$ N (Figure 7(h)). In correspondence of the low-level Eady growth rate increase, the upper-troposphere synoptic activity is intensified (see TEF in Figure 8(d)) and is associated, as in "TP+MP", with a southward shift and upstream intensification of the meridional eddy momentum convergence (purple contours in Figure 7(i)). This explains the strengthening and equatorward shift of the Pacific jet (green contours in Figure 8(d) and shading in Figure 4(o)). Hence, although the response to "TP" cooling is weaker in intensity compared to "TP+MP" cooling, surface forcing over the TP region is fundamental to obtain the environmental conditions that produce the atmospheric patterns in the latter experiment. The inclusion of MP cooling then reinforces the circulation anomalies in the western Pacific (see e.g. TEF and zonal-wind anomalies in Figure 8(b–d)).

# 4  Conclusions

By comparing a selection of CMIP6 historical simulations - the "cold Tibetan Plateau (TP) composite" - with an idealised AGCM simulation, we show how cold temperatures over Central Asia orography influence the winter atmospheric circulation over East Asia and the North Pacific. Colder than average Asian high plateaux strengthen the tropospheric heat sink and the East Asian winter monsoon, corresponding to an intensification of the north-westerly winds and of the downstream cold temperature advection. Over the East China Sea, the enhancement of the advection of cold northerly air from the continent

and of the surface heat flux from the ocean contribute to the intensification of the low-level baroclinicity. The results of the idealised experiment show that low-level baroclinic conditions over the East China Sea favour the development of transient atmospheric perturbations which deposit additional eddy momentum on the mean zonal flow, reinforcing the jet stream mainly upstream of the Pacific basin and on its equatorward flank (Hoskins et al., 1983; Hoskins and Valdes, 1990).

We note that the cooling of Central Asia orography interestingly corresponds to an overall amplification of the response to the uplift of the orography itself, presented in the works by Shi et al. (2015); Sha et al. (2015); White et al. (2017). This is in line with the results of the highly idealised study by Ringler and Cook (1999), which shows how the atmospheric response to simple patterns of orographic forcing is amplified (nonlinearly) by superposed cooling.

Building on previous literature that investigates the relative role of the Tibetan and Mongolian Plateaux on the downstream winter climate by removing or adding regional orography (Shi et al., 2015; Sha et al., 2015; White et al., 2017), we apply a similar approach to surface temperature forcing. In an additional set of idealised simulations, cold anomalies are confined to the regions of the Mongolian or of the Tibetan Plateau. The response to Tibetan Plateau cooling only, shows strong resemblance with the response to the total cooling pattern, supporting the fact that the TP region is fundamental for setting atmospheric conditions ideal for the intensification of the East Asian winter monsoon and of the Pacific jet, as in the CMIP6 models contributing to the "cold TP composite". The response to Mongolian Plateau cooling still consists in a strengthening of the zonal winds over the Pacific and reinforces the atmospheric response to Tibetan Plateau cooling. However, due to weakened advection of cold air to the east of the Tibetan Plateau, the jet intensification is shifted northward with respect to experiments with TP or total surface cooling. We note that a limited superposition of the two regional forcing patterns is present, due to a latitudinal smoothing of the anomalies in the TP forcing experiment.

The influence of East Asian surface temperature anomalies on the climate downstream is particularly relevant in the context of climate modelling, since state-of-the-art models are often affected by a cold surface and near-surface temperature bias over East Asia (Wei et al., 2014; Gong et al., 2014), which is accentuated over the Tibetan Plateau region (Peng et al., 2022; Fan et al., 2020, Figure 1,). Limited improvements have been detected, despite the model developments of the recent years (e.g. across CMIP phases, Bock et al., 2020; Lun et al., 2021; Hu et al., 2022). The issue is analysed in considerable detail by Chen et al. (2017), who decompose the surface energy budget over the TP and show that the processes causing surface and low-level cold biases are physically interlinked, and involve snow cover (and surface albedo), low-level water vapor content, downward longwave and shortwave radiation. The anomalies in the low-level heat fluxes ultimately result in a cooling of the boundary layer.

The results of this work suggests that thermal conditions over high Central Asia plateaux foster significant changes in the large-scale circulation on the lee side of the orography. Relating this to the cold Tibetan Plateau temperature bias measured across many climate models, it is possible to assert that such a surface anomaly potentially produces atmospheric biases over East Asia and the western North Pacific. Specifically, models characterised by colder-than-average temperatures over Central Asian plateaux present a strengthening of the East Asian winter monsoon, affecting the atmospheric conditions of the highly inhabited eastern coast of China and the Pacific jet. Although not considered in this work, the results also provide a new perspective on elevation dependent warming (EDW), implying that a stronger warming of Asian orography with respect to

other land regions may be important not only for the local climate, but also for the mean atmospheric conditions downstream. Further work is needed to assess such an impact of EDW.

Finally, based on the findings here presented, we prospect that advances in the representation of surface processes over complex orography will improve the modelling of the mean climate downstream of the Asian high plateaux and the inter-model spread in this region, with possible impacts on the confidence of regional multi-model climate projections. Otherwise, within the state of the art of model ensembles (e.g. CMIP6), the "emergent costraints" approach (Hall et al., 2019), applied to the feedback between surface temperatures over orography and the local energy budget, can become a useful means of reducing present uncertainty in East Asian climate projections. On a different time scale, works analysing subseasonal-to-seasonal forecasts over East Asia find a significant influence by surface anomalies over the Tibetan Plateau (e.g. Li et al., 2018; Xue et al., 2021), implying that shorter-term operational forecasting could also benefit from advances in the modelling of land–atmosphere interaction over Central Asia plateaux.

*Data availability.* The CMIP6 dataset is publicly available at https://esgf-node.llnl.gov/projects/cmip6/. Download information on the AGCM "SPEEDY" can be found at the link https://www.ictp.it/research/esp/models/speedy.aspx.

*Author contributions.* All authors conceived the study and contributed to the interpretation and discussion of the results. A. P. performed the analyses and wrote the paper.

*Competing interests.* No competing interests are present.

*Acknowledgements.* A. P. expresses her gratitude to Gwendal Rivière for insightful discussion and advice. The authors are also thankful to three anonymous reviewers whose comments and suggestions have contributed to improving the quality of the paper. Funding from Università Milano - Bicocca through project FAQC 2020-ATESP-0003 is acknowledged.

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

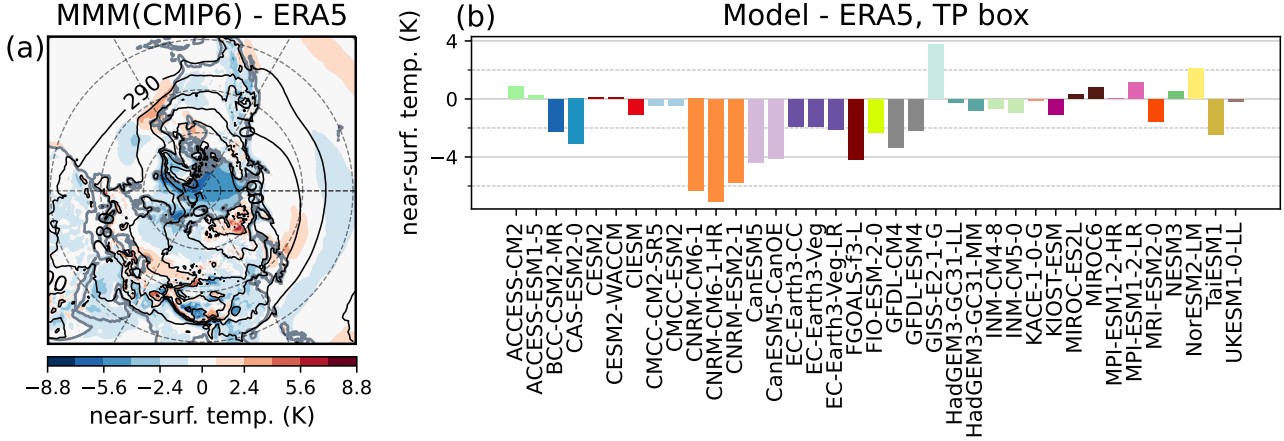

**Figure 1.** (a) The multi-model mean bias with respect to ERA5 in the Jan-Feb near-surface temperature climatology 1979–2008, with the ERA5 climatology in contours, and (b) the individual model biases over the TP box [25-40° N, 70-105° E] (see black box in panel (b) of Figure 2)

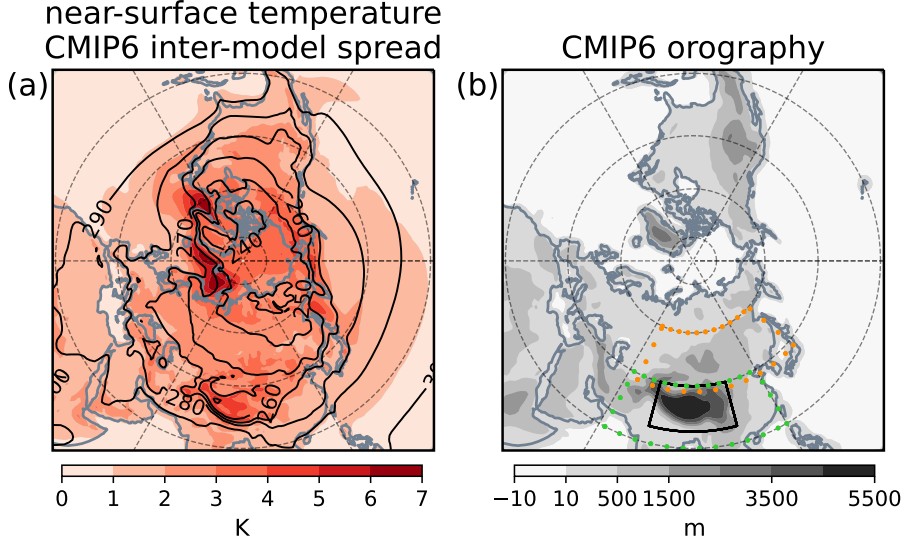

**Figure 2.** (a) The inter-model spread (standard deviation) in the Jan-Feb near-surface temperature climatology for CMIP6 historical 1979–2008 simulations, with the MMM field in contours, and (b) the MMM orographic elevation. The black longitude-latitude contour in panel (b), of range [25-40° N, 70-105° E], is the TP box used to compute the Tibetan Plateau index for near-surface temperature; the model biases in Figure 1(b) and the "cold TP composite" presented in Figures 3–6 are based on such index. The dotted boxes in panel (b) indicate the mountainous regions here named Tibetan Plateau or TP region (green) and Mongolian Plateau or MP region (orange)

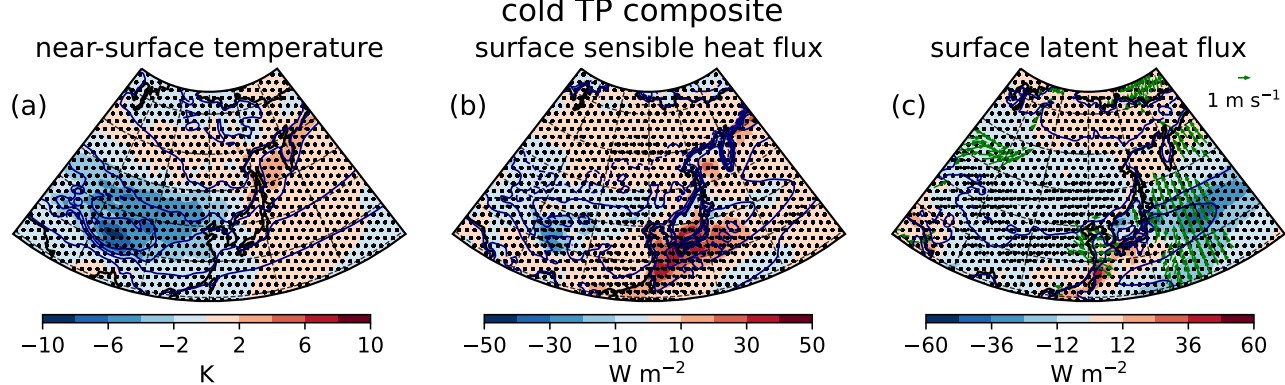

**Figure 3.** From the "cold TP composite" the anomalies of (a) near-surface temperature, (b) sensible and (c) latent surface heat flux (upward) and 1000 hPa horizontal wind vector (green arrows). Stippling and arrows indicate where anomalies exceed the 95th percentile in a randomly extracted 6-model composite distribution, see Methods. The respective MMM climatologies are displayed in contours (cl=[$\pm$5,+25,+50,+100,+200] W m$^{-2}$ for sensible heat flux, cl=[0,+10,+100,+200,+400] W m$^{-2}$ for latent heat flux)

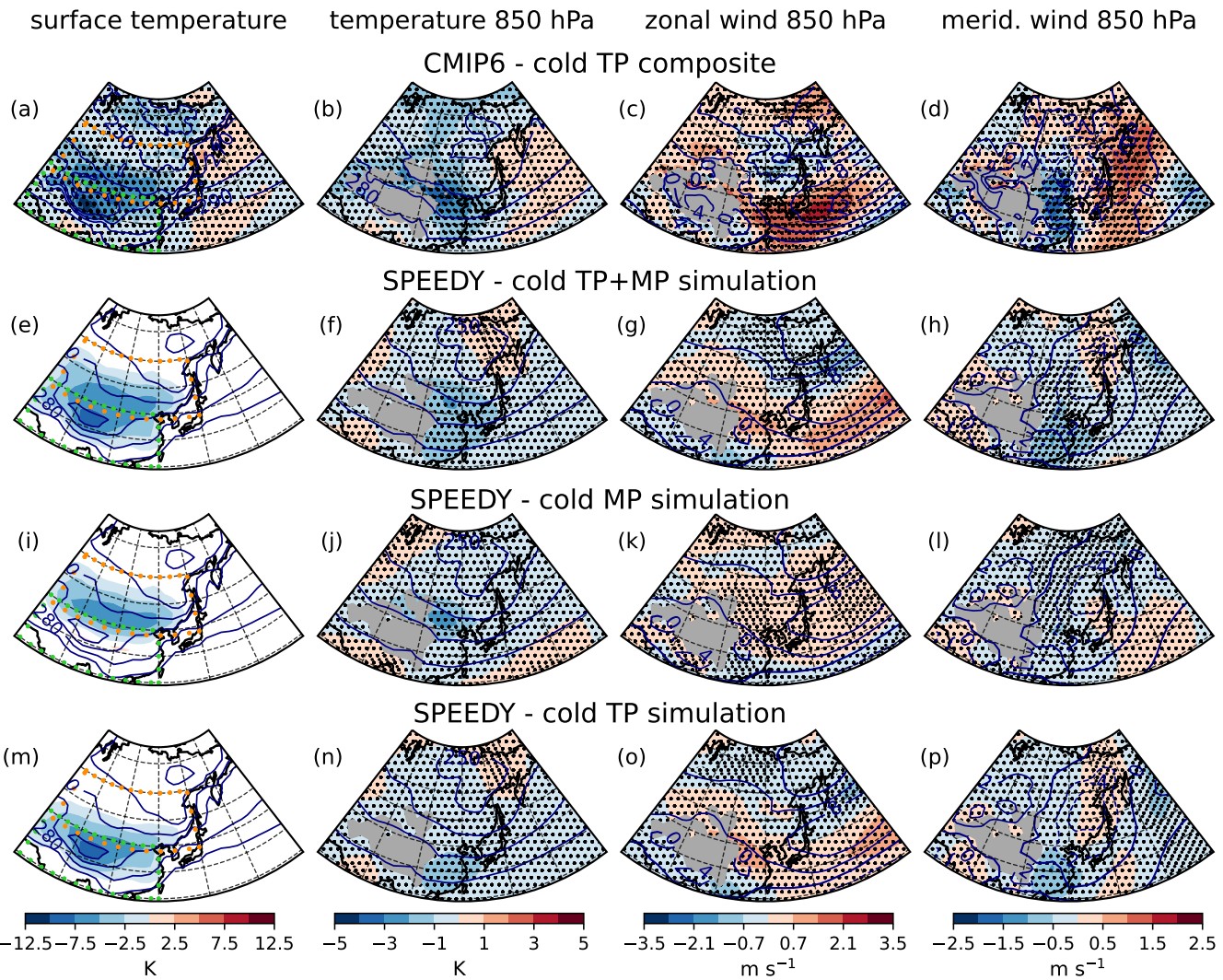

**Figure 4.** The "cold TP composite" anomalies of (a) surface temperature and 850-hPa (b) air temperature, (c) zonal wind, (d) meridional wind; the respective MMM climatologies in contours. The response of the model SPEEDY to "TP+MP", "MP" and "TP" surface-temperature forcing (panels (e,i,m)) in terms of 850-hPa (f,j,n) temperature, (g,k,o) zonal wind, (h,l,p) meridional wind; the control run in contours. Stippling shows the anomalies exceeding the 95th percentile of a randomly extracted distribution (see Methods). Green and orange dotted boxes in panels (a,e,i,m) indicate the mountainous areas named TP region and MP region, respectively. Grey shading masks orography exceeding 1400 m

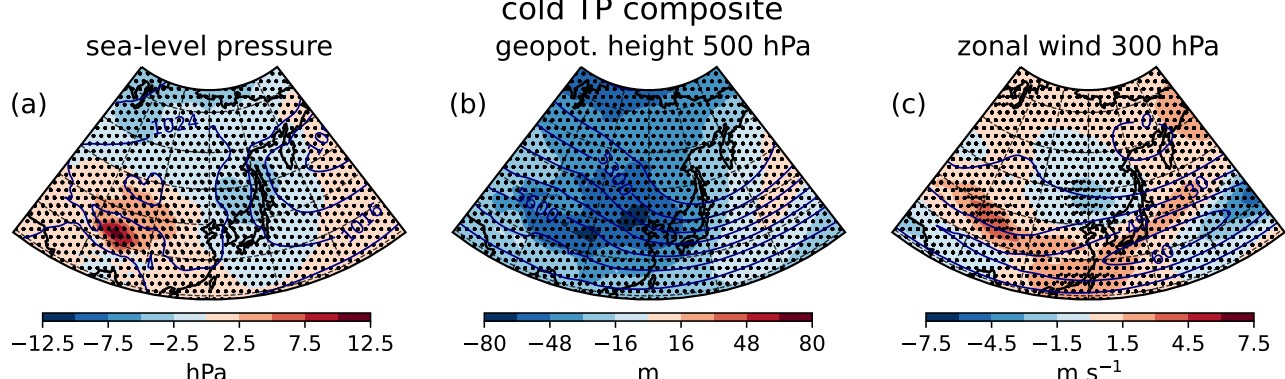

**Figure 5.** The "cold TP composite" anomalies of (a) sea-level pressure, (b) 500 hPa geopotential height and (c) 300 hPa zonal wind; the respective MMM climatologies in contours. Stippling shows the anomalies exceeding the 95th percentile in a randomly extracted 6-model composite distribution, see Methods

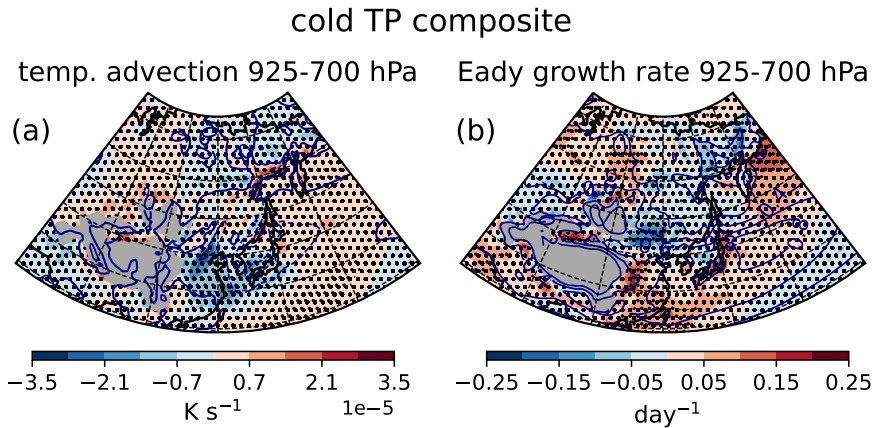

**Figure 6.** The "cold TP composite" anomalies of (a) temperature advection by the mean flow ($\boldsymbol{u} \cdot \nabla T$) averaged over the pressure-levels 925 to 700 hPa, (b) Eady growth rate between 925 and 700 hPa, and the respective MMM climatologies in contours (ci=4e-5 $\mathrm{K\,s^{-1}}$ for temperature advection, stippling for anomalies exceeding the 95th percentile in a randomly extracted 6-model composite distribution, see Methods). Grey shading masks orography exceeding 1400 m

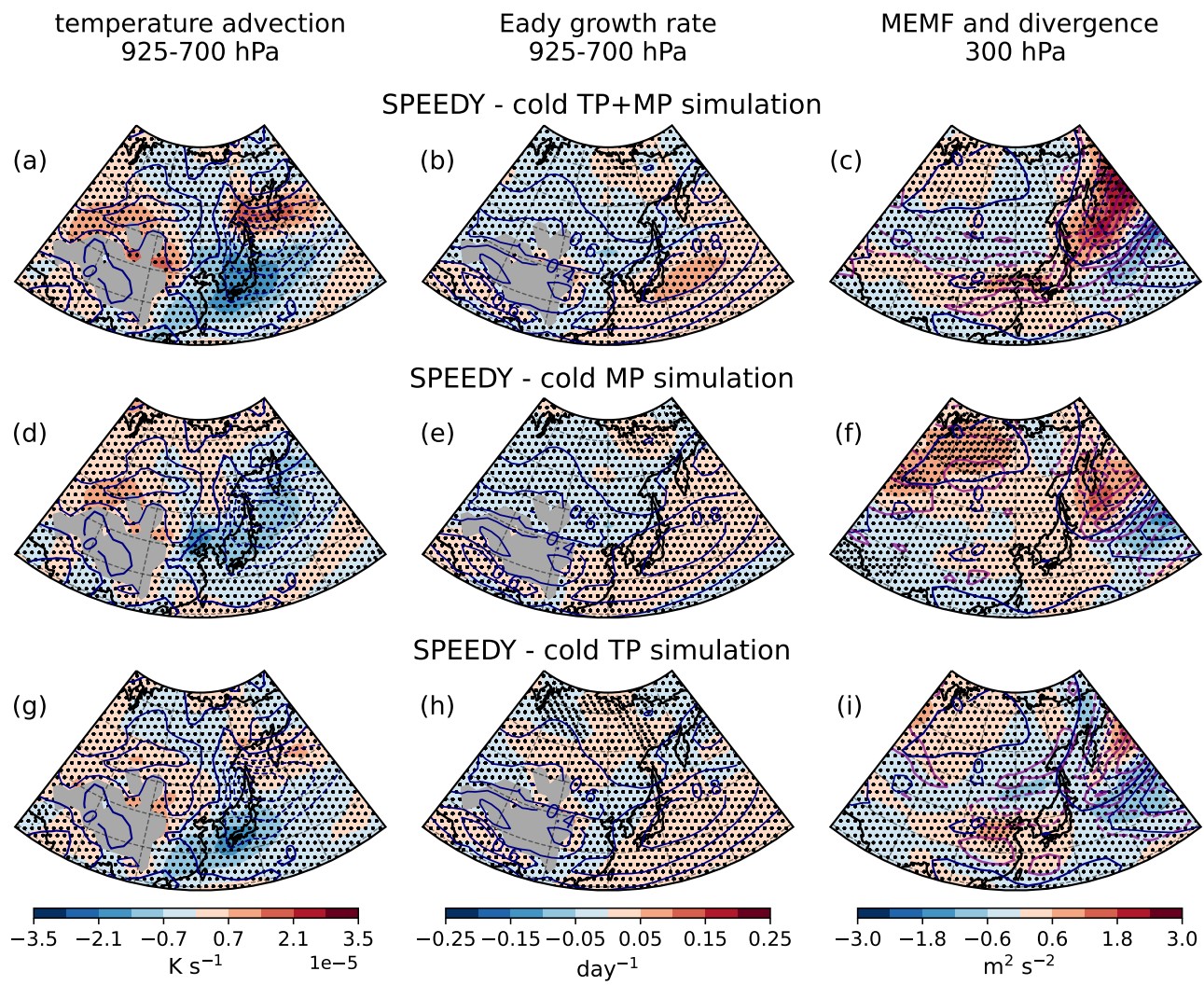

**Figure 7.** The response of the model SPEEDY to "TP+MP", "MP" and "TP" surface-temperature forcing in terms of (a,d,g) temperature advection by the mean flow ($\boldsymbol{u} \cdot \nabla T$) averaged over the pressure levels 925 to 700 hPa, (b,e,h) Eady growth rate between 925 and 700 hPa, (c,f,i) meridional eddy momentum flux (MEMF) at 300 hPa and its divergence (in purple contours for cl=[$\pm$5,$\pm$15,+25]e-7 m s$^{-2}$). The control run is shown in contours (ci=4e-5 K s$^{-1}$ for temperature advection, ci=5 m$^2$ s$^{-2}$ for MEMF) and stippling indicates where the anomalies exceed the 95th percentile of a randomly extracted distribution (see Methods). Grey shading masks orography exceeding 1400 m

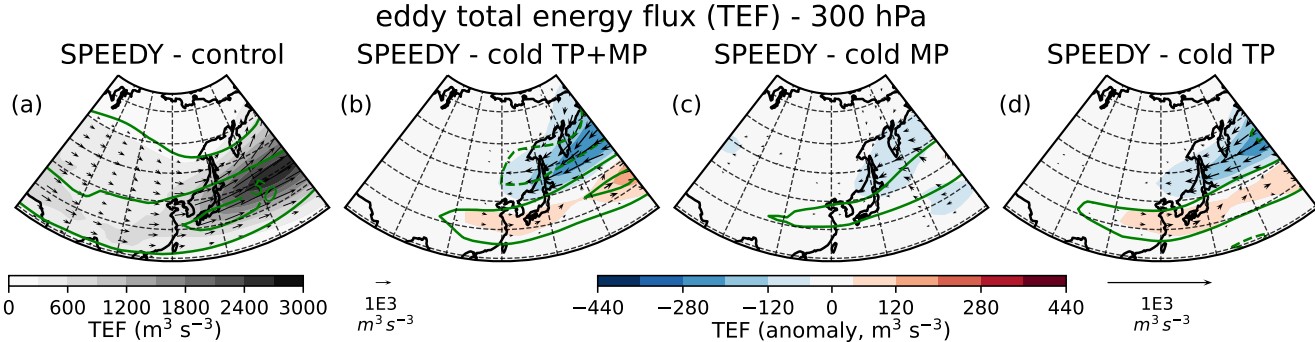

**Figure 8.** (a) The 300 hPa eddy total energy flux (TEF) climatology in the SPEEDY control integration, and the TEF response to (b) "TP+MP", (c) "MP" and (d) "TP" surface-temperature forcing. The zonal wind is shown in green contours (ci=20 m s$^{-1}$ for the control climatology in panel (a), cl=[±1,±3] m s$^{-1}$ for the response in panels (b–d))

**Table 1.** List of CMIP6 climate models

| Model Name | Member Id. | Institution | Horizontal Resolution (lon × lat) |
|---|---|---|---|
| ACCESS-CM2 | 1 | Australian Research Council Centre of Excellence for Climate System Science & Commonwealth Scientific and Industrial Research Organisation (AUS) | 1.9° × 1.3° |
| ACCESS-ESM1-5 | 1 | Commonwealth Scientific and Industrial Research Organisation (AUS) | 1.9° × 1.2° |
| BCC-CSM2-MR | 1 | Beijing Climate Center (CHN) | 1.1° × 1.1° |
| **CanESM5** | 1 | Canadian Centre for Climate Modelling and Analysis (CAN) | 2.8° × 2.8° |
| **CanESM5-CanOE** | 1 | as above | 1.9° × 1.9° |
| CAS-ESM2-0 | 2 | Chinese Academy of Sciences (CHN) | 1.4° × 1.4° |
| CESM2 | 2 | National Center for Atmospheric Research, Climate and Global Dynamics Laboratory (USA) | 1.3° × 0.9° |
| CESM2-WACCM | 1 | as above | 1.3° × 0.9° |
| CIESM | 1 | Department of Earth System Science, Tsinghua University (CHN) | 0.9° × 1.3° |
| CMCC-CM2-SR5 | 1 | Fondazione Centro Euro-Mediterraneo sui Cambiamenti Climatici (ITA) | 0.9° × 1.3° |
| CMCC-ESM2 | 1 | as above | 0.9° × 1.3° |
| **CNRM-CM6-1** | 1 | Centre National de Recherches Meteorologiques & Centre Européen de Récherche et de Formation Avancée en Calcul Scientifique (FRA) | 1.4° × 1.4° |
| **CNRM-CM6-1-HR** | 1 | as above | 0.5° × 0.5° |
| **CNRM-ESM2-1** | 1 | as above | 1.4° × 1.4° |
| EC-Earth3-CC | 1 | EC-Earth consortium (visit https://ec-earth.org/consortium/) | 0.7° × 0.7° |
| EC-Earth3-Veg | 1 | as above | 0.7° × 0.7° |
| EC-Earth3-Veg-LR | 1 | as above | 1.1° × 1.1° |
| **FGOALS-f3-L** | 1 | Chinese Academy of Sciences (CHN) | 1.3° × 1° |
| FIO-ESM-2-0 | 1 | Qingdao National Laboratory for Marine Science and Technology & First Institute of Oceanography (CHN) | 1.3° × 0.9° |
| GFDL-CM4 | 1 | National Oceanic and Atmospheric Administration, Geophysical Fluid Dynamics Laboratory (USA) | 1.3° × 1° |
| GFDL-ESM4 | 1 | as above | 1.3° × 1° |
| GISS-E2-1-G | 1 | Goddard Institute for Space Studies (USA) | 2.5° × 2° |
| HadGEM3-GC31-LL | 1 | Met Office Hadley Centre (GBR) | 1.9° × 1.2° |
| HadGEM3-GC31-MM | 1 | as above | 0.8° × 0.6° |
| INM-CM4-8 | 1 | Institute for Numerical Mathematics (RUS) | 2° × 1.5° |
| INM-CM5-0 | 1 | as above | 2° × 1.5° |
| KACE-1-0-G | 1 | National Institute of Meteorological Sciences/Korea Meteorological Administration (KOR) | 1.3° × 0.9° |
| KIOST-ESM | 1 | Korea Institute of Ocean Science & Technology (KOR) | 1.9° × 1.9° |
| MIROC6 | 1 | as above | 1.4° × 1.4° |
| MIROC-ES2L | 1 | Japan Agency for Marine-Earth Science and Technology & Atmosphere and Ocean Research Institute & National Institute for Environmental Studies & RIKEN Center for Computational Science (JPN) | 2.8° × 2.8° |
| MPI-ESM1-2-HR | 1 | Max Planck Institute for Meteorology (DEU) | 0.9° × 0.9° |
| MPI-ESM1-2-LR | 1 | as above | 1.9° × 1.9° |
| MRI-ESM2-0 | 1 | Meteorological Research Institute (JPN) | 1.1° × 1.1° |
| NESM3 | 1 | Nanjing University of Information Science and Technology (CHN) | 1.9° × 1.9° |
| NorESM2-LM | 1 | NorESM Climate modeling Consortium (visit https://www.noresm.org/consortium/) | 2.5° × 1.9° |
| TaiESM1 | 1 | Research Center for Environmental Changes (TWN) | 0.9° × 1.3° |
| UKESM2-0-LL | 1 | National Institute of Meteorological Sciences/Korea Meteorological Administration (KOR) | 1.9° × 1.3° |

Models in bold were selected for the "cold Tibetan-Plateau" composite