# Peer review of "Atmospheric response to cold wintertime Tibetan Plateau conditions over East Asia in climate models"

_EGUsphere, 2022_

## Referee Comment (RC2)

**Review of "Atmospheric response to wintertime Tibetan Plateau cold bias in climate models" by Portal et al.**

The authors present an interesting and sound analysis of the influence of cold biases in CMIP6 models/AGCM and its influence on the atmospheric state across east Asia and the North Pacific. The results presented by the authors is of interest and relevant to the journal, however I believe it could be improved by expanding on some results further and exploring model sensitivity. Furthermore, a greater explanation of the results from the SPEEDY model would be welcome. I list my several major points below as well as some more minor comments. Once these are addressed I see no reason why this manuscript should not be accepted for publication

**Major Comments**

1. It would be beneficial to show some of the model spread in the cold bias. Several things that would improve the analysis are: is it only the cold models that have the downstream response in heat fluxes, wind biases, eady growth rate, etc. A comparison of warm and cold models would be useful. Furthermore, all changes are expressed relative to the model mean, but are the models already biased relative to the observations/reanalysis? Do these cold models amplify an existing model mean bias or how much of the bias can be associated with the colder models? Do the coldest models have the largest biases in heat fluxes etc? I suggest plotting a scatter plot of temperature bias in the TP/MP region against average heat flux (or wind bias) downstream to test this.
2. What is the spread in response in the SPEEDY simulations? No stippling is shown in Fig. 5. I suggest something similar as above to investigate the variability in AGCM response.
3. You have performed a TP+MP and an MP cold experiments and come to the conclusion that most of the downstream response is a result of the TP forcing. Surely running experiments of just the TP cold bias would answer this question. I suggest the authors address this in some way.

**Minor Comments**

1. L37-38: I suggest adding a reference to Fig. 1b here.
2. L150: hyphen required in years.
3. L151 and Fig. 1a: how do you determine spread? Is this just the standard deviation of the temperature at each grid point?
4. L154: incorrect colour labelling and figure reference – please correct.
5. L167: 'land' not required.
6. L180-185: suggest adding more explanation here on the mechanism as to how the cold TP bias influences the flow downstream. This will just need to add some discussion from the introduction I believe.
7. L188-189: suggest adding some lat/lon co-ordinates to reference which part of the Chinese coast line you are referring to – it's slightly confusing.
8. Is there anything particular about the models that have the largest cold bias? Are they of lowest horizontal or vertical resolution?

---

## Author Comment (AC1)

Based on the composites of CMIP6 models, this paper shows a cold bias over the Tibetan Plateau (TP), and finds that the negative temperature anomalies over TP intensify the East Asia winter monsoon by enhancing the low-level baroclinicity in the region of the East China Sea. Then, the southern flank of the Pacific jet is reinforced. The responses of AGCM experiments support the results of CMIP6 composite. Results are interesting and a cause for concern. The manuscript is generally well written and the methods appear sound. Since there are still some points need to be revised, I would like to recommend a moderate revision before this paper can be accepted for publication.

**General comments:**

1.1 Why exclude December from the analysis? The period of December-January-February is usually considered as the deep winter in East Asia, and December should be included in order to assess the full climatology of TP temperature and Pacific jet. In addition, for the climate conditions in Asian region, the January-February sometimes denotes the late-winter. The temperature variability and atmospheric climatology associated with the East Asian winter monsoon have obviously subseasonal variations (e.g., Zhong & Wu, 2022, https://doi.org/10.1007/s00382-022-06610-9; Park & Kim, 2021, https://doi.org/10.1007/s00382-020-05544-4; Tian & Fan, 2020, https://doi.org/10.1007/s00382-019-05068-6) from the early- to late-winter. So, if the cold TP bias and the related dynamic processes proposed in the study are also applicable in the early-winter (i.e., November-December)?

We agree with reviewer's point, but our analysis showed that the "cold-TP composite" anomalies in temperature and 850-hPa zonal wind (as shown by supporting Figures R1 and R2 here included) based on the periods "December-January-February" and "January-February" are almost indistinguishable. Because of this, because of easier access to SPEEDY experiments, and finally for consistency with the related Portal et al. 2022 (ref. in the manuscript, using a similar experimental setup), we prefer to show the results in terms of January-February winters. Moreover, the choice of JF is not uncommon within the topic, e.g. see

Jhun, J., & Lee, E. (2004). A New East Asian Winter Monsoon Index and Associated Characteristics of the Winter Monsoon, *Journal of Climate*, *17*(4), 711-726.

Clark, M. P. and Serreze, M. C.: Effects of variations in East Asian snow cover on modulating atmospheric circulation over the North Pacific Ocean, Journal of Climate, 13, 3700–3710, 2000.

In Methods 2.1 we add the following sentence:

*"The outcome is equivalent for December-January-February winters."*

[Figure]

Figure R1: near-surface temperature anomaly in the "cold TP composite", version JF (left) and DJF (right). Figure on the left is as Figure 2a from the manuscript.

[Figure]

Figure R2: 850-hPa zonal-wind anomaly in the "cold TP composite", version JF (left) and DJF (right). Figure on the left is as Figure 2c from the manuscript

1.2 Some model evaluations for the SPEEDY are needed, and at least one to be considered is the climatology of East Asian jet in AGCM and in CMIP6. Although the authors use the same surface temperature forcing as those in TP composite to drive the AGCM, the strength and position of cold advection and eddy growth rate are different (Figures 4 & 5). Compared to the CMIP6 composite, the temperature advection and eddy growth rate in AGCM are distributed farther east and closer to the Pacific and may contribute to less climate effects over the East Asian continent. Is the climatology of Asian jet in SPEEDY different from that in CMIP6 MME?

We thank Reviewer #1 for the interesting comment. Indeed, the 850 hPa zonal wind of the SPEEDY climatology and of the CMIP6 MMM are considerably different (see Figure R3 below): this is expected since SPEEDY is not a full-fledged atmospheric climate model, but relies on a simplified physics. Indeed, the eddy-driven jet in SPEEDY is weaker and shifted towards the north of the Pacific basin. However, in both cases the TP cooling affects the 850 hPa jet similarly over the East-Asian coast, with a weakening of the winds to the east of the orography. The limited eastward extension of the positive signal in the MMM (Figures 3(c) and 4(b)) is probably linked to other factors influencing the Pacific circulation within the cold TP composite (see e.g. the positive temperature advection in Figure 4(a) and surface latent heat flux Figure 2(c)) more than to differences in the mean state. We will mention this shortcoming of the composite analysis in the Results when drafting the new version of the manuscript.

[Figure]

Figure R3: difference in Jan-Feb 850-hPa zonal wind between SPEEDY climatology and the CMIP6 multi-model mean.

1.3 How does the cold TP bias construct in CMIP6 climate modes, snow cover over TP or other processes related to the surface heat fluxes? More discussions should be provided in the manuscript. Additionally, for the temperature advection, the authors only present the cold

advection by the mean flow (i.e., Figure 4a). However, the anomalous low-level winds are also important to advect the surface temperature. How about the temperature advection by the anomalous winds?

The following paragraph will be inserted in the Introduction to provide a plausible explanation for the TP cold bias in terms of heat fluxes.

*"The CMIP5 cold Tibetan Plateau temperature bias has been examined in some detail by Chen et al. (2017). Among the climate models taking part in CMIP5 they identify a strong bias in the western region of the Plateau and show that it is more evident in terms of near-surface than surface (skin) temperature. The reason for the emergence of the strong near-surface bias is investigated by decomposing the different contributions to the low-level energy budget. Anomalous snow cover corresponds to an increase in the surface albedo, hence in the reflection of shortwave radiation, and this is anti-correlated with upward turbulent heat fluxes. While the surface temperature is weakly affected by these terms, due to compensation between incoming shortwave radiative and outgoing turbulent fluxes, a reduction in the turbulent heat flux into the atmosphere, leading to a decrease in the low-level water vapor content and thermal radiation, cools the boundary layer. By identifying physically interlinked low-level and surface processes modifying the energy budget, Chen et al. (2017) are able to explain why many CMIP5 models present a low-level cold bias over the Tibetan Plateau. These findings are likely applicable to the CMIP6 models, which show similar biases over the TP."*

The temperature advection ($\mathbf{u} \cdot \nabla T$) is the dot-product between the velocity and the gradient vectors. Its anomaly from the climatological value takes into account the "cold composite" climatological anomalies from the MMM wind and temperature fields. This means that the temperature advection has not been expressed in terms of decomposition of the two linear terms (i.e. anomalous temperature advection by the MMM mean flow and MMM temperature advection by the anomalous flow). We hope this answer addresses the second part of the Reviewer's comment.

1.4 In fact, the cold bias of winter temperature is not limited to TP, but the whole East Asia which is similar to those in CMIP5 models (e.g., Gong et al., 2014, https://doi.org/10.1175/JCLI-D-13-00039.1; Wei et al., 2014, https://doi.org/10.1007/s00382-013-1929-z). I suggest that the authors should give a brief discussion about the cold bias between CMIP5 and CMIP6 models.

We extend in the Introduction the discussion of East-Asia temperature biases throughout the recent CMIP phases following the advice of the Reviewer. The suggested references will be also mentioned in the Conclusions. Here the modification that will be inserted:

*"An additional motivation to approach this topic is the presence of a significant multi-model mean (MMM) temperature bias in the East Asia region, which is evident over successive phases of the CMIP and over multiple seasons. Priestley et al. (2022) detect a strong deviation from the reanalysis for summer temperature and, based on the modified thermal gradients in the low troposphere, hypothesise a role of the TP land temperature on the baroclinicity and cyclogenesis downstream. Along the same lines, East-Asia winter conditions are anomalously cold among several climate models (Wei et al., 2014; Gong et al., 2014), although improvements, associated with a closer representation of the winter monsoon, have been detected in the transition from CMIP Phase 3 to Phase 5 (Wei et al., 2014). The winter bias is especially strong over the TP (Peng et al., 2022; Fan et al., 2020, and Figure 1). Limited progress is obtained in transition from CMIP5 to CMIP6 (Lun et al., 2021; Hu et al., 2022). These studies also highlight the presence of a wide inter-model spread in year-round East-Asia and TP temperatures among the CMIP climate models,*

*which appears to be related with the difficulties in representing surface energy fluxes (Wei et al., 2014), in particular over regions characterised by complex orography and seasonal variations in snow cover (e.g. Su et al., 2013; Chen et al., 2017; Li et al., 2021)."*

Furthermore, to better frame the role of the TP bias, and to provide more support to this discussion, the following figure, showing the Northern Hemisphere MMM near-surface temperature bias (left), and for each model the bias over the TP (right), will also be included in the updated manuscript.

[Figure]

**Figure 1.** (a) The multi-model mean bias with respect to ERA5 in the Jan-Feb near-surface temperature climatology 1979–2008, with the ERA5 climatology in contours, and (b) the individual model biases over the TP box [25-40 N, 70-105 E] (see black box in panel (b) of Figure 2)

**Specific comments:**

2.1 Suggest to change the title to reflect the East Asian winter monsoon. How about "Atmospheric responses in East Asia to wintertime Tibetan Plateau cold bias in CMIP6 models"?

Following the reviewer's comment, the title has been changed to: "Atmospheric response to cold wintertime Tibetan Plateau conditions over East Asia in climate models".

2.2 The physical processes associated with the atmospheric responses to the cold TP bias are in line with expectations and previous analyses as illustrated in Introduction (i.e., L42-L47). So the novelty of the study needs to be better explained given these works.

We add some context to the statement of novelty of the study. We state now that the cold TP forcing amplifies the atmospheric response to East-Asian orographic forcing shown by previous works (modifications in Abstract, Results, Conclusions). As an example, we report the first paragraph of the Conclusions:

*"By comparing a selection of CMIP6 historical simulations - the "cold Tibetan Plateau (TP) composite" - with an idealised AGCM simulation we show how cold temperatures over Central Asia orography influence the winter atmospheric circulation over East Asia and the North Pacific. Colder than average Asian plateaux strengthen the tropospheric heat sink and intensify the East Asia winter monsoon, leading to stronger north-westerly winds and cold advection downstream of the orographic features. The response to cooling of Central Asia orography also coincides with an overall amplification of the response to mechanical forcing from the same orography (Shi et al., 2015; Sha et al., 2015; White et al., 2017). Moreover,*

*this is in line with the idealised study by Ringler and Cook (1999), combining simple patterns of orographic (mechanic) and thermal forcing."*

2.3 Throughout the paper, words like the "bias" and "spread" are cross-used. They should have different meanings, and it's better to define them more clearly in the paper.

We thank the reviewer for pointing this out. In the new version of the manuscript, we address this problem by using the two terms more accurately. This is helped by the addition of a new figure (Figure 1 above) displaying CMIP6 near surface temperature bias.

2.4 "Cold bias" in your title should mean the temperature difference between the model simulation and observation. However, no observation data are used and the definition of "cold TP composite" in the paper does not meet the meaning of "bias". I suggest a more appropriate word.

See answer 2.3. The word bias has been removed from the title.

2.5 In Table 1, it would be more helpful to provide the latitude & longitude resolution in degrees or grid cells for each model.

The longitude x latitude grid spacing is inserted in Table 1.

2.6 L187: Figure 2(b, d) may be Figure 3(b, d).

We thank the reviewer for the correction.

---

## Author Comment (AC2)

The authors present an interesting and sound analysis of the influence of cold biases in CMIP6 models/AGCM and its influence on the atmospheric state across east Asia and the North Pacific. The results presented by the authors is of interest and relevant to the journal, however I believe it could be improved by expanding on some results further and exploring model sensitivity. Furthermore, a greater explanation of the results from the SPEEDY model would be welcome. I list my several major points below as well as some more minor comments. Once these are addressed I see no reason why this manuscript should not be accepted for publication.

**Major Comments**

1.1 It would be beneficial to show some of the model spread in the cold bias. Several things that would improve the analysis are: is it only the cold models that have the downstream response in heat fluxes, wind biases, Eady growth rate, etc. A comparison of warm and cold models would be useful. Furthermore, all changes are expressed relative to the model mean, but are the models already biased relative to the observations/reanalysis? Do these cold models amplify an existing model mean bias or how much of the bias can be associated with the colder models? Do the coldest models have the largest biases in heat fluxes etc? I suggest plotting a scatter plot of temperature bias in the TP/MP region against average heat flux (or wind bias) downstream to test this.

The role of the CMIP6 TP cold bias has been further expanded in the new version of the manuscript. A new figure (Figure 1 in the revised manuscript, see below) now shows the bias in near-surface temperature, completing the information on the inter-model spread (Figure 1a in the former version of the manuscript). We have corrected the text in order to avoid confusion between the terms bias, spread and anomaly from the multi-model mean.

[Figure]

**Figure 1.** (a) The multi-model mean bias with respect to ERA5 in the Jan-Feb near-surface temperature climatology 1979–2008, with the ERA5 climatology in contours, and (b) the individual model biases over the TP box [25-40 N, 70-105 E] (see black box in panel (b) of Figure 2)

Furthermore, in Figure R4 (below) we report a scatter plot of TP near-surface temperature against surface sensible heat flux in a lon-lat box to the south of Japan. It shows a rather linear relation between the near-surface temperature on the Tibetan Plateau and the downstream sensible heat flux (uw.) - with correlation coefficient of -.0.85, confirming that the TP temperature plays a role in the downstream conditions generally among CMIP6 models, not only in the "cold TP composite". We report this new result in the manuscript as in the following. Note that the figure numbers refer to the revised version of the manuscript (to be submitted).

*"The strong surface heat flux anomalies present over the Pacific basin in the "cold TP composite" (Figure 3(b,c)) are related to the strengthening of the Pacific jet over and downstream of the East China Sea (Figure 4(c)), which extend down to the near-surface level (green arrows in Figure 3(c)) and intensify the advection of cold air masses over the ocean (Figure 6(a)). Indeed, cold air temperatures and strong winds in the boundary layer reinforce the surface turbulent heat fluxes by the warmer sea surface. We note that the relation between (i) cold TP temperatures, (ii) strong low-level winds entering the Pacific basin south of Japan and (iii) strong sensible heat fluxes from the ocean surface over the South China Sea, shows a linear tendency across the CMIP6 models (e.g. the correlation coefficient between (i) and (iii) is -0.85, where (i) is the near-surface temperature in the TP box (black box in Figure 3(b)) and (iii) is the surface sensible heat flux in a [25-40 N, 120-135 E] box). This confirms that the impact of the TP thermal state on the dynamical features over East Asia is not just a peculiarity of the "cold TP composite", but rather extends to the whole CMIP6 ensemble. "*

[Figure]

Figure R4: Scatter plot between TP near surface temperature (area-weighted average over lat-lon box [25-40N,70-105N]) and Pacific surface sensible heat fluxes south of Japan (area-weighted average over lat-lon box [25-40N,120-135N]).

1.2 What is the spread in response in the SPEEDY simulations? No stippling is shown in Fig. 5. I suggest something similar as above to investigate the variability in AGCM response.

The significance of the variables in SPEEDY has been computed and stippling has been added to Figure 5 (will correspond to Figure 7 in the revised manuscript).

1.3 You have performed a TP+MP and MP cold experiments and come to the conclusion that most of the downstream response is a result of the TP forcing. Surely running experiments of just the TP cold bias would answer this question. I suggest the authors address this in some way.

We are grateful to the reviewer for this suggestion, which elucidates further how the temperature in the TP region leads to a southward shift in the downstream circulation. We added the results in the figures (new figures 4,7) and expanded the corresponding discussion in the Results and Conclusions sections.

**Minor Comments**

2.1 L37-38: I suggest adding a reference to Fig. 1b here.

The reference has been included.

2.2 L150: hyphen required in years.

The typo has been corrected.

2.3 L151 and Fig. 1a: how do you determine spread? Is this just the standard deviation of the temperature at each grid point?

We thank the reviewer for pointing out the missing information. In the Results section and in the caption it is added that the spread is the standard deviation of the models' climatologies.

2.4 L154: incorrect colour labelling and figure reference – please correct.

Correction applied.

2.5 L167: 'land' not required.

Correction applied.

2.6 L180-185: suggest adding more explanation here on the mechanism as to how the cold TP bias influences the flow downstream. This will just need to add some discussion from the introduction I believe.

Some more detailed perspectives on the reasons for jet strengthening are given in the paragraph following, that will be part of the new Results section. Please note that the Figures will have modified numbering in the new manuscript.

*"The advection of cold air downstream of the TP (Figure 6(a), see Methods for details on the computation) is supported both by the negative temperature anomaly on the orography and, to the east, by the reinforcement of the north-westerly wind (Figure 4(b,d)). These conditions are responsible for intensified meridional temperature gradients east of the TP and along the Pacific coast which enhance the baroclinicity (see positive anomalies in the Eady growth rate west and east of the chinese coastline at latitudes 20–40 N, Figure 6(b)). Given that the Eady growth rate (definition in Methods) measures the environmental conditions favourable to atmospheric baroclinic instability, we would expect the strengthening of the jet at the entrance of the Pacific basin (Figure 4(c)): this should be induced by increased synoptic*

*activity east of the TP and over the East China Sea, a region where cyclogenesis is climatologically high in mid winter (Priestley et al., 2020; Schemm et al., 2021). However, an analysis of the eddy feedback on the zonal flow for the idealised "TP+MP experiment" - generally coherent with the results of the CMIP6 composite analysis - supports the hypothesis that the jet strengthening is induced by a decline in the synoptic activity to the north of the jet maximum, rather than by an increased activity to its south, as prospected by the Eady growth rate east of the Chinese coast. This will be discussed in more detail in the description of the idealised experiments."*

2.7 L188-189: suggest adding some lat/lon co-ordinates to reference which part of the Chinese coast line you are referring to – it's slightly confusing.

Latitude reference has been included.

2.8 Is there anything particular about the models that have the largest cold bias? Are they of lowest horizontal or vertical resolution?

Information about models' resolution has been included in Table 1, but no evident correlation exists between cold TP and resolution, suggesting that this might be part of the land-surface scheme.

---

## Author Comment (AC3)

This paper investigated the atmospheric response to wintertime cold Tibetan Plateau (TP) bias with CMIP6 multi-model mean (MMM) simulations and idealized SPEEDY experiments. The authors found that the cold bias over Asian orography intensifies the East Asia winter monsoon (EAWM) through enhancing the low-level baroclinity and reinforcing the southern Pacific jet. The EAWM is a three-dimensional climate system and more details should be examined to measure its strength. Thus, I recommend a major and mandatory revision before the paper could be accepted. The details of the comments are listed below.

**Major comments:**

This study investigated the impacts of the cold bias over Asian orography on East Asia winter monsoon (EAWM). The EAWM is a three-dimensional climate system (e.g., Jhun et al. 2004) and its strength could not be simply measured by the wind at 850hPa. Thus, the authors should carefully check the atmospheric anomaly (e.g., Z500, U300, SLP) to measure the strength of EAWM (Jhun et al. 2004, Wang et al. 2010).

We agree with the reviewer's comment. A new figure (Figure 5 below) detailing the vertical structure of the EAWM, is added to the manuscript in order to address the reviewers' concern. The consistency with East Asia winter monsoon strengthening, with reference to Jhun et al. 2004, is discussed in the Results as in the following:

*"Typical features relatable to a strong East Asia winter monsoon are captured by the sea-level pressure and mid-troposphere geopotential height fields in Figure 5, as by comparing e.g. with strong and weak monsoon conditions in Figure 6 of Jhun and Lee (2004). A deeper zonal pressure contrast to the east of the Siberian High (panel (a)) and a lower 500 hPa isobaric surface over the Asian coast (panel (b)), together with a strong 300 hPa jet inland and to the south of Japan (panel (c)) - following the low-level signal (Figure 4(c)), are in conformity with maps describing the atmospheric state associated with an intense monsoon."*

[Figure]

**Figure 5.** The "cold TP composite" anomalies of (a) sea-level pressure, (b) 500 hPa geopotential height and (c) 300 hPa zonal wind; the respective MMM climatologies in contours. Stippling shows the anomalies exceeding the 95th percentile in a randomly extracted 6-model composite distribution, see Methods

Other comments:

2.1 Line 101: 'the January and February months are referred to as winter'. Why December is not considered? In general, the boreal winter is referred as "December-January- February" (DJF).

The results of the model compositing do not change if considering JF instead of DJF. See Figures R1 and R2 included below. This will be mentioned in the Methods section of the new manuscript.

[Figure]

Figure R1: near-surface temperature anomaly in the "cold TP composite", version JF (left) and DJF (right). Figure on the left is as Figure 2a from the manuscript.

[Figure]

Figure R2: 850-hPa zonal-wind anomaly in the "cold TP composite", version JF (left) and DJF (right). Figure on the left is as Figure 2c from the manuscript

2.2 Line 37: 40°N could be better.

Correction applied.

2.3 Line 128: As mentioned in line 125, the LST is prescribed in SPEEDY model. However, it is also proposed that the model includes a freely evolving LST scheme. I wonder how the LST is treated in the SPEEDY simulations? Could the LST be affected by upper-level circulation, or it is just prescribed as a model input? Please clarify.

The LST in the freely evolving versions interacts with the upper-level circulation by responding to a surface energy-balance equation, while relaxing towards a prescribed climatology - the relaxation time scale is 40 days. In the prescribed LST case (used here) the LST is simply fixed to the prescribed field, and does not evolve.

A detailed description of the LST scheme is available in Appendix B of

Portal, A., C. Pasquero, F. D'Andrea, P. Davini, M. E. Hamouda, and G. Rivière, 2022: Influence of Reduced Winter Land–Sea Contrast on the Midlatitude Atmospheric Circulation. *J. Climate*, **35**, 2637–2651, https://doi.org/10.1175/JCLI-D-21-0941.1.

Such reference is provided also in the Methods of the manuscript, section 2.2.

2.4 Line 150: '1979-2008' could be better.

The correction has been applied.

2.5 Figure 2: Please check the unit of the heat flux. It could be W m-2.

Correct, we thank the reviewer for pointing out this mistake.

2.6 Figure 2: Positive value means upward or downward heat flux? Please provide the information in figure captions.

The issue has been addressed.

2.7 Line 170: The statement could be misleading. The heat flux change is negative over TP regions.

We specify in the description (Results) that enhanced cooling is present where the turbulent heat fluxes are climatologically negative.

2.8 Line 171: If the heat flux change is not significant over TP and CP, why the authors show the heat flux change here? It could confuse the readers.

From Figure 2b,c (old manuscript numbering) we note that stippling (significant signal) is present over the TP and MP, even if it does not extend uniformly over the region.

2.9 Line 184: The jet stream distributes around 300hPa during winter (Jhun et al. 2004). The statement here could be misleading.

We specify that in this context we are referring to the eddy-driven jet.

2.10 Line 187: Please check the figure captions.

The caption has been corrected.

2.11 Line 190: Increased instability favors acceleration of upper-level zonal winds (e.g., Nie et al., 2016). Please show the zonal wind change of upper troposphere.

The upper-level jet is shown in the new Figure 5(c), shown in response to the Reviewer's major comment.

2.12 Line 214: Please check the figure captions.

The caption has been revised in order to make it clearer.

2.13 Figure 5: Please show the significant information of the changes as in Figure 4.

Stippling to indicate the  significant changes has been included in the updated figure.

2.14 Line 199: Please show the surface wind anomaly with vectors. Otherwise, one may not understand the heat flux anomaly.

Following the reviewer's comments, we have inserted in Figure 2c (old numbering) vectors showing significant wind anomalies at 1000 hPa (near-surface wind was unavailable for ~10 models, hence the 1000 hpa pressure level was preferred).

2.15 Line 200: More upward heat flux? Please clarify.

The sentence has been rephrased.

Reference:

Wang, B., Wu, Z., Chang, C., Liu, J., Li, J., & Zhou, T. (2010). Another Look at Interannual-to-Interdecadal Variations of the East Asian Winter Monsoon: The Northern and Southern Temperature Modes, Journal of Climate, 23(6), 1495-1512.

---

## Author Response (AR1)

27th April 2023

Dear Editor,

We are thankful for the possibility of reviewing our work for the EGU journal *Weather and Climate Dynamics.*

Some substantial modifications have been added with respect to the initial pre-print of the manuscript. While in-text revisions are provided in the marked-up version of the manuscript, we list some major changes in the points below. Please note that figures are referred to by their number in the manuscript revision.

- Following comment 1.3 by Reviewer #2, a new idealised experiment run with the SPEEDY model, featuring a Tibetan Plateau (TP) cooling only, was added to the previous Mongolian Plateau (MP) and MP+TP experiments. New panels relative to the TP experiment are now shown in **Figures 4, 7** and **8** and discussed in the **Results** section.

- Three new Figures and a revision of **Figure 7** are inserted and described in the manuscript to address issues raised by the Reviewers (for details see point-by point answer below).

    - **Figure 1** shows the CMIP6 multi-model mean bias and the individual models' biases over the TP in terms of near-surface temperature.

    - **Figure 5** displays the mean sea-level pressure, 500 hPa geopotential height and 300 hPa zonal wind fields, which are useful to assert the strengthening of the East Asia winter monsoon.

    - **Figure 8** provides information on the upper-level eddy total energy flux (TEF). It is useful for understanding the propagation and intensity of the eddy energy over the Pacific and allows an intuitive interpretation of the changes in MEMF (RHS of Figure 7). The significance analysis is not applied to the results of this Figure.

    - The RHS panels of **Figure 7** have been revised to account for the significance analysis and for upper-level (instead of lower-level) changes in the synoptic variability. It now shows meridional eddy momentum flux at a pressure level of 300 hP (MEMF, significant where stippling is shown) and its convergence (purple contours).

We have addressed all the Reviewers' comments and acknowledge major improvements in the manuscript thanks to the recommended changes. In the following we include the point-by-point answer to the comments and the marked-up version of the manuscript.

Kind regards,

Alice Portal (on behalf of the authors)

Note that line and figure numbers refer to the marked-up manuscript.

**Reviewer #1**

Based on the composites of CMIP6 models, this paper shows a cold bias over the Tibetan Plateau (TP), and finds that the negative temperature anomalies over TP intensify the East Asia winter monsoon by enhancing the low-level baroclinicity in the region of the East China Sea. Then, the southern flank of the Pacific jet is reinforced. The responses of AGCM experiments support the results of CMIP6 composite. Results are interesting and a cause for concern. The manuscript is generally well written and the methods appear sound. Since there are still some points need to be revised, I would like to recommend a moderate revision before this paper can be accepted for publication.

**General comments:**

1.1 Why exclude December from the analysis? The period of December-January-February is usually considered as the deep winter in East Asia, and December should be included in order to assess the full climatology of TP temperature and Pacific jet. In addition, for the climate conditions in Asian region, the January-February sometimes denotes the late-winter. The temperature variability and atmospheric climatology associated with the East Asian winter monsoon have obviously subseasonal variations (e.g., Zhong & Wu, 2022, https://doi.org/10.1007/s00382-022-06610-9; Park & Kim, 2021, https://doi.org/10.1007/s00382-020-05544-4; Tian & Fan, 2020, https://doi.org/10.1007/s00382-019-05068-6) from the early- to late-winter. So, if the cold TP bias and the related dynamic processes proposed in the study are also applicable in the early-winter (i.e., November-December)?

We agree with the reviewer's point. However, our analysis shows that the "cold-TP composite" anomalies in temperature and 850-hPa zonal wind (as shown by supporting Figures R1 and R2, here included) based on the periods "December-January-February" and "January-February" are almost indistinguishable. Because of this, and for consistency with the related Portal et al. 2022 (ref in the manuscript, using a similar experimental setup), we prefer to show the results in terms of January-February winters. Moreover, the choice of JF is not uncommon within the topic, e.g. see

Jhun, J., & Lee, E. (2004). A New East Asian Winter Monsoon Index and Associated Characteristics of the Winter Monsoon, *Journal of Climate*, *17*(4), 711-726.

Clark, M. P. and Serreze, M. C.: Effects of variations in East Asian snow cover on modulating atmospheric circulation over the North Pacific Ocean, Journal of Climate, 13, 3700–3710, 2000.

We mention the similarity of JF and DJF results in Methods 2.1, **line 136**.

[Figure]

Figure R1: near-surface temperature anomaly in the "cold TP composite", version JF (left) and DJF (right). Figure on the left is as Figure 2a from the manuscript.

[Figure]

Figure R2: 850-hPa zonal-wind anomaly in the "cold TP composite", version JF (left) and DJF (right). Figure on the left is as Figure 2c from the manuscript

1.2 Some model evaluations for the SPEEDY are needed, and at least one to be considered is the climatology of East Asian jet in AGCM and in CMIP6. Although the authors use the same surface temperature forcing as those in TP composite to drive the AGCM, the strength and position of cold advection and eddy growth rate are different (Figures 4 & 5). Compared to the CMIP6 composite, the temperature advection and eddy growth rate in AGCM are distributed farther east and closer to the Pacific and may contribute to less climate effects over the East Asian continent. Is the climatology of Asian jet in SPEEDY different from that in CMIP6 MME?

We thank Reviewer #1 for the interesting comment. Indeed, the 850 hPa zonal wind of the SPEEDY climatology and of the CMIP6 MMM are considerably different (see Figure R3 below): this is expected since SPEEDY is not a full-fledged atmospheric climate model, but relies on a simplified physics. Indeed, the eddy-driven jet in SPEEDY is weaker and shifted towards the north of the Pacific basin. However, in both cases the TP cooling affects the 850 hPa jet similarly over the East-Asian coast, with a weakening of the winds to the east of the orography. The limited eastward extension of the positive signal in the MMM (Figures 3(c) and 4(b)) is probably linked to other factors influencing the Pacific circulation within the cold TP composite (see e.g. the positive temperature advection in Figure 4(a) and surface latent heat flux Figure 2(c)) more than to differences in the mean state. We mention this shortcoming in the Results, **line 325**.

[Figure]

Figure R3: difference in Jan-Feb 850-hPa zonal wind between SPEEDY climatology and the CMIP6 multi-model mean (MMM). Contours denote the MMM isolines.

1.3 How does the cold TP bias construct in CMIP6 climate modes, snow cover over TP or other processes related to the surface heat fluxes? More discussions should be provided in the manuscript. Additionally, for the temperature advection, the authors only present the cold advection by the mean flow (i.e., Figure 4a). However, the anomalous low-level winds are also important to advect the surface temperature. How about the temperature advection by the anomalous winds?

The paragraph in **lines 104-116** has been inserted in the Introduction to provide a plausible explanation for the TP cold bias in terms of heat fluxes.

The temperature advection ($\mathbf{u} \cdot \nabla T$) is the dot-product between the velocity and the gradient vectors. Its anomaly from the climatological value takes into account the "cold composite" climatological anomalies from the MMM wind and temperature fields. This means that the temperature advection has not been expressed in terms of decomposition of the two linear terms (i.e. anomalous temperature advection by the MMM mean flow and MMM temperature advection by the anomalous flow).

1.4 In fact, the cold bias of winter temperature is not limited to TP, but the whole East Asia which is similar to those in CMIP5 models (e.g., Gong et al., 2014, https://doi.org/10.1175/JCLI-D-13-00039.1; Wei et al., 2014, https://doi.org/10.1007/s00382-013-1929-z). I suggest that the authors should give a brief discussion about the cold bias between CMIP5 and CMIP6 models.

We extend in the Introduction the discussion of East-Asia temperature biases throughout the recent CMIP phases following the advice of the Reviewer (**lines 90-99**). The suggested references are also mentioned in the Conclusions (**line 407**).

Furthermore, to better frame the discussion on the TP bias, **Figure 1**, showing the Northern Hemisphere MMM near-surface temperature bias (**a**) and the bias over the TP for each model (**b**), is also included in the revision.

**Specific comments:**

2.1 Suggest to change the title to reflect the East Asian winter monsoon. How about "Atmospheric responses in East Asia to wintertime Tibetan Plateau cold bias in CMIP6 models"?

Following the reviewer's comment, the title has been changed to: "Atmospheric response to cold wintertime Tibetan Plateau conditions over East Asia in climate models".

2.2 The physical processes associated with the atmospheric responses to the cold TP bias are in line with expectations and previous analyses as illustrated in Introduction (i.e., L42-L47). So the novelty of the study needs to be better explained given these works.

We add context to the statement of novelty of the study. We state that the cold TP forcing amplifies the atmospheric response to East-Asian orographic forcing shown by previous works (modifications in Abstract - **line 14**, Results - **lines 268-272**, Conclusions - **lines 386-389**).

2.3 Throughout the paper, words like the "bias" and "spread" are cross-used. They should have different meanings, and it's better to define them more clearly in the paper.

We thank the reviewer for pointing this out. In the new version of the manuscript, we address this problem by using the two terms more accurately. This is helped by the addition of a **Figure 1** displaying CMIP6 near surface temperature bias.

2.4 "Cold bias" in your title should mean the temperature difference between the model simulation and observation. However, no observation data are used and the definition of "cold TP composite" in the paper does not meet the meaning of "bias". I suggest a more appropriate word.

See answer to comment 2.3. The word "bias" has been removed from the title.

2.5 In Table 1, it would be more helpful to provide the latitude & longitude resolution in degrees or grid cells for each model.

The longitude x latitude grid spacing has been inserted in **Table 1**.

2.6 L187: Figure 2(b, d) may be Figure 3(b, d).

We thank the reviewer for the correction.

**Reviewer #2**

The authors present an interesting and sound analysis of the influence of cold biases in CMIP6 models/AGCM and its influence on the atmospheric state across east Asia and the North Pacific. The results presented by the authors is of interest and relevant to the journal, however I believe it could be improved by expanding on some results further and exploring model sensitivity. Furthermore, a greater explanation of the results from the SPEEDY model would be welcome. I list my several major points below as well as some more minor comments. Once these are addressed I see no reason why this manuscript should not be accepted for publication.

**Major Comments**

1.1 It would be beneficial to show some of the model spread in the cold bias. Several things that would improve the analysis are: is it only the cold models that have the downstream response in heat fluxes, wind biases, Eady growth rate, etc. A comparison of warm and cold models would be useful. Furthermore, all changes are expressed relative to the model mean, but are the models already biased relative to the observations/reanalysis? Do these cold models amplify an existing model mean bias or how much of the bias can be associated with the colder models? Do the coldest models have the largest biases in heat fluxes etc? I suggest plotting a scatter plot of temperature bias in the TP/MP region against average heat flux (or wind bias) downstream to test this.

The role of the CMIP6 TP cold bias has been further expanded in the new version of the manuscript. **Figure 1** now shows the bias in near-surface temperature, completing the information on the inter-model spread shown in **Figure 2a**. We have corrected the text in order to avoid confusion between the terms bias, spread and anomaly from the multi-model mean.

Furthermore, in Figure R4 (below) we report a scatter plot of TP near-surface temperature against surface sensible heat flux in a lon-lat box to the south of Japan. It shows a substantially linear relation between the near-surface temperature on the Tibetan Plateau and the downstream sensible heat flux (uw.), with correlation coefficient of -.0.85. confirming that the TP temperature plays a role in the downstream conditions generally among CMIP6 models, not only in the "cold TP composite". We report this new result in the manuscript at **lines 294-301**.

[Figure]

Figure R4: Scatter plot between TP near surface temperature (area-weighted average over lat-lon box [25-40N,70-105N]) and Pacific surface sensible heat fluxes south of Japan (area-weighted average over lat-lon box [25-40N,120-135N]).

1.2 What is the spread in response in the SPEEDY simulations? No stippling is shown in Fig. 5. I suggest something similar as above to investigate the variability in AGCM response.

The significance of the variables in SPEEDY has been computed and stippling has been added to **Figure 7** (former Fig 5).

1.3 You have performed a TP+MP and MP cold experiments and come to the conclusion that most of the downstream response is a result of the TP forcing. Surely running experiments of just the TP cold bias would answer this question. I suggest the authors address this in some way.

We are grateful to the reviewer for this suggestion, which elucidates further how the temperature in the TP region leads to a strengthening and southward shift of the downstream circulation. We included the description of the new TP forcing experiment in the Methods and its results in **Figures 4,7,8**, and expanded the corresponding discussion in the Results (**lines 364-373**) and Conclusion (**lines 396-404**).

**Minor Comments**

2.1 L37-38: I suggest adding a reference to Fig. 1b here.
The reference has been included.

2.2 L150: hyphen required in years.
The typo has been corrected.

2.3 L151 and Fig. 1a: how do you determine spread? Is this just the standard deviation of the temperature at each grid point?

We thank the reviewer for pointing out the missing information. In the Results section (**line 217**) and in the caption of **Figure 2**, we include text explaining that the spread is the standard deviation over the models' climatologies.

2.4 L154: incorrect colour labelling and figure reference – please   correct.

Correction applied.

2.5 L167: 'land' not required.

Correction applied.

2.6 L180-185: suggest adding more explanation here on the mechanism as to how the cold TP bias influences the flow downstream. This will just need to add some discussion from the introduction I believe.

We expand on the reasons for jet strengthening in the paragraph at **lines 273-287**. Please note that the Figures have modified numbering in the new manuscript.

2.7 L188-189: suggest adding some lat/lon co-ordinates to reference which part of the Chinese coast line you are referring to – it's  slightly confusing.

Latitude reference has been included.

2.8 Is there anything particular about the models that have the largest cold bias? Are they of lowest horizontal or vertical resolution?

Information about models' resolution has been included in **Table 1**, but no evident correlation exists between cold TP and resolution, suggesting that this might be part of the land-surface scheme.

**Anonymous Referee #3**

This paper investigated the atmospheric response to wintertime cold Tibetan Plateau (TP) bias with CMIP6 multi-model mean (MMM) simulations and idealized SPEEDY experiments. The authors found that the cold bias over Asian orography intensifies the East Asia winter monsoon (EAWM) through enhancing the low-level baroclinity and reinforcing the southern Pacific jet. The EAWM is a three-dimensional climate system and more details should be examined to measure its strength. Thus, I recommend a major and mandatory revision before the paper could be accepted. The details of the comments are listed below.

**Major comments:**

This study investigated the impacts of the cold bias over Asian orography on East Asia winter monsoon (EAWM). The EAWM is a three-dimensional climate system (e.g., Jhun et al. 2004) and its strength could not be simply measured by the wind at 850hPa. Thus, the authors should carefully check the atmospheric anomaly (e.g., Z500, U300, SLP) to measure the strength of EAWM (Jhun et al. 2004, Wang et al. 2010).

We agree with the reviewer's comment. **Figure 5** is added to the manuscript in order to detail the vertical structure of the EAWM. The consistency with East Asia winter monsoon strengthening, with reference to Jhun et al. 2004, is discussed in the Results **lines 264-272**. Moreover, in the SPEEDY experiments, we add in **Figures 7,8** and in the paragraphs from **line 330** to **line 373** results regarding the upper-level storm track and its momentum deposition on the tropospheric jet.

**Other comments:**

2.1 Line 101: 'the January and February months are referred to as winter'. Why December is not considered? In general, the boreal winter is referred as "December-January- February" (DJF).

The results of the model compositing do not change if considering JF instead of DJF. See Figures R1 and R2 included below. This is mentioned in the Methods section, **line 136**.

[Figure]

Figure R1: near-surface temperature anomaly in the "cold TP composite", version JF (left) and DJF (right). Figure on the left is as Figure 2a from the manuscript.

[Figure]

Figure R2: 850-hPa zonal-wind anomaly in the "cold TP composite", version JF (left) and DJF (right). Figure on the left is as Figure 2c from the manuscript

2.2 Line 37: 40°N could be better.

Correction applied.

2.3 Line 128: As mentioned in line 125, the LST is prescribed in SPEEDY model. However, it is also proposed that the model includes a freely evolving LST scheme. I wonder how the LST is treated in the SPEEDY simulations? Could the LST be affected by upper-level circulation, or it is just prescribed as a model input? Please clarify.

The LST in the freely evolving versions interacts with the upper-level circulation by responding to a surface energy-balance equation, while relaxing towards a prescribed climatology - the relaxation time scale is 40 days. In the prescribed LST case (used here) the LST is simply fixed to the prescribed field, and does not evolve.

A detailed description of the LST scheme is available in Appendix B of

Portal, A., C. Pasquero, F. D'Andrea, P. Davini, M. E. Hamouda, and G. Rivière, 2022: Influence of Reduced Winter Land–Sea Contrast on the Midlatitude Atmospheric Circulation. *J. Climate*, **35**, 2637–2651, https://doi.org/10.1175/JCLI-D-21-0941.1.

Such reference is provided also in the Methods of the manuscript, section 2.2.

2.4 Line 150: '1979-2008' could be better.

The correction has been applied.

2.5 Figure 2: Please check the unit of the heat flux. It could be W m-2.

Correct, we thank the reviewer for pointing out this mistake.

2.6 Figure 2: Positive value means upward or downward heat flux? Please provide the information in figure captions.

It is an upward heat flux. The issue has been addressed.

2.7 Line 170: The statement could be misleading. The heat flux change is negative over TP regions.

We specify in the description (Results) that enhanced cooling is present where the turbulent heat fluxes are climatologically negative, **lines 240-242**.

2.8 Line 171: If the heat flux change is not significant over TP and CP, why the authors show the heat flux change here? It could confuse the readers.

From **Figure 3b,c** we note that stippling (significant signal) is present over the TP and the MP, even if it does not extend uniformly over the entire region. We clarify the confusion in **lines 242-243**.

2.9 Line 184: The jet stream distributes around 300hPa during winter (Jhun et al. 2004). The statement here could be misleading.

We now specify that in this context we are referring to the eddy-driven jet.

2.10 Line 187: Please check the figure captions.

The caption has been corrected.

2.11 Line 190: Increased instability favors acceleration of upper-level zonal winds (e.g., Nie et al., 2016). Please show the zonal wind change of upper troposphere.

The upper-level jet is now shown in the **Figure 5(c)** and the analysis of eddy momentum deposition in the SPEEDY experiment is also shown for the upper troposphere (**Figures 7,8**).

2.12 Line 214: Please check the figure captions.

The caption has been revised.

2.13 Figure 5: Please show the significant information of the changes as in Figure 4.

Stippling to indicate the  significant changes has been included in the updated figure.

2.14 Line 199: Please show the surface wind anomaly with vectors. Otherwise, one may not understand the heat flux anomaly.

Following the reviewer's comments, we have inserted in **Figure 3c** vectors showing significant wind anomalies at 1000 hPa (near-surface wind was unavailable for ~10 models, hence the 1000 hpa pressure level was preferred).

2.15 Line 200: More upward heat flux? Please clarify.

The sentence has been rephrased.

Reference:

Wang, B., Wu, Z., Chang, C., Liu, J., Li, J., & Zhou, T. (2010). Another Look at Interannual-to-Interdecadal Variations of the East Asian Winter Monsoon: The Northern and Southern Temperature Modes, Journal of Climate, 23(6), 1495-1512.

[revised manuscript text omitted]

near-surface temperature
CMIP6 inter-model spread

(a)

CMIP6 orography

(b)

0  1  2  3  4  5  6  7
K

−10 10 500 1500  3500  5500
m

**Figure 2.** (a) The inter-model spread (standard deviation) in the Jan-Feb near-surface temperature climatology for CMIP6 historical 1979–2008 simulations, with the MMM field in contours, and (b) the MMM orographic elevation. The black longitude-latitude contour in panel (b), of range [25-40 N, 70-105 E], is the TP box used to compute the Tibetan Plateau index for near-surface temperature; the model biases in Figure 1(b) and the "cold TP composite" presented in Figures 3–6 are based on such index. The dotted boxes in panel (b) indicate the mountainous regions here named Tibetan Plateau or TP region (green) and Mongolian Plateau or MP region (orange)

[Figure]

cold TP composite

near-surface temperature

(a)

surface sensible heat flux

(b)

surface latent heat flux

(c)

1 m s⁻¹

−10  −6  −2  2  6  10
K

−50 −30 −10 10  30  50
W m⁻²

−60 −36 −12 12  36  60
W m⁻²

**Figure 3.** From the "cold TP composite" the anomalies of (a) near-surface temperature, (b) sensible and (c) latent surface heat flux (upward) and 1000 hPa horizontal wind vector (green arrows). Stippling and arrows indicate where anomalies exceed the 95th percentile in a randomly extracted 6-model composite distribution, see Methods. The respective MMM climatologies are displayed in contours (cl=[±5,+25,+50,+100,+200] W m⁻² for sensible heat flux, cl=[0,+10,+100,+200,+400] W m⁻² for  latent heat flux)

[Figure]

**Figure 4.** The "cold TP composite" anomalies of (a) surface temperature and 850-hPa (b) air temperature, (c) zonal wind, (d) meridional wind.  ; the respective MMM climatologies  in contours. The response of the model SPEEDY to "TP+MP" , "MP" and "TP" surface-temperature forcing (panels (e,i,m)) in terms of 850-hPa (f,j,n) temperature, (g,k,o) zonal wind, (h,l,p) meridional wind; the control run  in contours. Stippling  shows the anomalies  exceeding the  95th percentile of a randomly extracted distribution (see Methods). Green and orange dotted boxes in panels (a,e,i,m) indicate the mountainous  areas named TP region and MP region, respectively. Grey shading masks orography exceeding 1400 m

cold TP composite

[Figure]

**Figure 5.** The "cold TP composite" anomalies of (a) sea-level pressure, (b) 500 hPa geopotential height and (c) 300 hPa zonal wind; the respective MMM climatologies in contours. Stippling shows the anomalies exceeding the 95th percentile in a randomly extracted 6-model composite distribution, see Methods

cold TP composite

[Figure]

**Figure 6.** The "cold TP composite" anomalies of (a) temperature advection by the mean flow ($\boldsymbol{u} \cdot \nabla T$) averaged over the pressure-levels 925 to 700 hPa, (b) Eady growth rate between 925 and 700 hPa, and the respective MMM climatologies in contours (ci=4e-5 $\mathrm{K\,s}^{-1}$ for temperature advection, stippling  for anomalies exceeding the  95th percentile in a randomly extracted 6-model composite distribution, see Methods). Grey shading masks orography exceeding 1400 m

[Figure]

**Figure 7.** The response of the model SPEEDY to "TP+MP", "MP" and "TP" surface-temperature forcing in terms of (a,d,g) temperature advection by the mean flow ($\boldsymbol{u} \cdot \nabla T$) averaged over the pressure levels 925 to 700 hPa, (b,e,h) Eady growth rate between 925 and 700 hPa, (c,f,i)  meridional eddy momentum flux (MEMF) at  300 hPa  and its divergence (in purple contours for cl=[±5,±15,+25]e-7 m s$^{-2}$). The control run is shown in contours (ci=4e-5 K s$^{-1}$ for temperature advection, ci=5 m$^2$ s$^{-2}$ for MEMF) and stippling indicates where the anomalies exceed the 95th percentile of a randomly extracted distribution (see Methods). Grey shading masks orography exceeding 1400 m

[Figure]

**Figure 8.** (a) The 300 hPa eddy total energy flux (TEF) climatology in the SPEEDY control integration, and the TEF response to (b) "TP+MP", (c) "MP" and (d) "TP" surface-temperature forcing. The zonal wind is shown in green contours (ci=20 m s$^{-1}$ for the control climatology in panel (a), cl=[$\pm 1, \pm 3$] m s$^{-1}$ for the response in panels (b–d))

**Table 1.** List of CMIP6 climate models

| Model Name | Member Id. | Institution | Horizontal Resolution (lon × lat) |
|---|---|---|---|
| ACCESS-CM2 | 1 | Australian Research Council Centre of Excellence for Climate System Science & Commonwealth Scientific and Industrial Research Organisation (AUS) | $1.9° \times 1.3°$ |
| ACCESS-ESM1-5 | 1 | Commonwealth Scientific and Industrial Research Organisation (AUS) | $1.9° \times 1.2°$ |
| BCC-CSM2-MR | 1 | Beijing Climate Center (CHN) | $1.1° \times 1.1°$ |
| **CanESM5** | 1 | Canadian Centre for Climate Modelling and Analysis (CAN) | $2.8° \times 2.8°$ |
| **CanESM5-CanOE** | 1 | as above | $1.9° \times 1.9°$ |
| CAS-ESM2-0 | 2 | Chinese Academy of Sciences (CHN) | $1.4° \times 1.4°$ |
| CESM2 | 2 | National Center for Atmospheric Research, Climate and Global Dynamics Laboratory (USA) | $1.3° \times 0.9°$ |
| CESM2-WACCM | 1 | as above | $1.3° \times 0.9°$ |
| CIESM | 1 | Department of Earth System Science, Tsinghua University (CHN) | $0.9° \times 1.3°$ |
| CMCC-CM2-SR5 | 1 | Fondazione Centro Euro-Mediterraneo sui Cambiamenti Climatici (ITA) | $0.9° \times 1.3°$ |
| CMCC-ESM2 | 1 | as above | $0.9° \times 1.3°$ |
| **CNRM-CM6-1** | 1 | Centre National de Recherches Meteorologiques & Centre Européen de Récherche et de Formation Avancée en Calcul Scientifique (FRA) | $1.4° \times 1.4°$ |
| **CNRM-CM6-1-HR** | 1 | as above | $0.5° \times 0.5°$ |
| **CNRM-ESM2-1** | 1 | as above  |  |
|  EC-Earth3-CC | 1 | EC-Earth consortium (visit https://ec-earth.org/consortium/) | $0.7° \times 0.7°$ |
| EC-Earth3-Veg | 1 | as above | $0.7° \times 0.7°$ |
| EC-Earth3-Veg-LR | 1 | as above | $1.1° \times 1.1°$ |
| **FGOALS-f3-L** | 1 | Chinese Academy of Sciences (CHN) | $1.3° \times 1°$ |
| FIO-ESM-2-0 | 1 | Qingdao National Laboratory for Marine Science and Technology & First Institute of Oceanography (CHN) | $1.3° \times 0.9°$ |
| GFDL-CM4 | 1 | National Oceanic and Atmospheric Administration, Geophysical Fluid Dynamics Laboratory (USA) | $1.3° \times 1°$ |
| GFDL-ESM4 | 1 | as above | $1.3° \times 1°$ |
| GISS-E2-1-G | 1 | Goddard Institute for Space Studies (USA) | $2.5° \times 2°$ |
| HadGEM3-GC31-LL | 1 | Met Office Hadley Centre (GBR) | $1.9° \times 1.2°$ |
| HadGEM3-GC31-MM | 1 | as above | $0.8° \times 0.6°$ |
| INM-CM4-8 | 1 | Institute for Numerical Mathematics (RUS) | $2° \times 1.5°$ |
| INM-CM5-0 | 1 | as above | $2° \times 1.5°$ |
| KACE-1-0-G | 1 | National Institute of Meteorological Sciences/Korea Meteorological Administration (KOR) | $1.3° \times 0.9°$ |
| KIOST-ESM | 1 | Korea Institute of Ocean Science & Technology (KOR) | $1.9° \times 1.9°$ |
| MIROC6 | 1 | as above | $1.4° \times 1.4°$ |
| MIROC-ES2L | 1 | Japan Agency for Marine-Earth Science and Technology & Atmosphere and Ocean Research Institute &  $2.8°$ National Institute for Environmental Studies & RIKEN Center for Computational Science (JPN) | |
| MPI-ESM1-2-HR | 1 | Max Planck Institute for Meteorology (DEU) | $0.9° \times 0.9°$ |
| MPI-ESM1-2-LR | 1 | as above | $1.9° \times 1.9°$ |
| MRI-ESM2-0 | 1 | Meteorological Research Institute (JPN) | $1.1° \times 1.1°$ |
| NESM3 | 1 | Nanjing University of Information Science and Technology (CHN) | $1.9° \times 1.9°$ |
| NorESM2-LM | 1 | NorESM Climate modeling Consortium (visit https://www.noresm.org/consortium/) | $2.5° \times 1.9°$ |
| TaiESM1 | 1 | Research Center for Environmental Changes (TWN) | $0.9° \times 1.3°$ |
| UKESM2-0-LL | 1 | National Institute of Meteorological Sciences/Korea Meteorological Administration (KOR) | $1.9° \times 1.3°$ |

Models in bold were selected for the "cold Tibetan-Plateau" composite

---

## Author Response (AR3)

We are greatful to the Editor and Reviewer #2 for carefully reading the manuscript, and for reporting some mistakes and issues. As in the point-by-point answer here included, we have proceeded to clarify and correct the manuscript where suggested.
Please note that the line references are relative to the revised marked-up manuscript.

EDITOR

-Experiment setting of SPEEDY runs: Based on the description in line 165, the SPEEDY is run in perpetual-winter mode (200 January and 200 February months). However, in the description of "control integration" , it is said in line 170 that the model is run with "evolving SSTs 1979-2008 from HadISST". Thus it is not clear to me how the surface SST is set for the perpetual-winter mode (200 January and 200 February months) in the SPEEDY experiments.
Please note that « the LST of the control experiment corresponds to the climatology obtained from a SPEEDY 10-member ensemble, run with a freely evolving LST scheme and with prescribed climatological SIC and evolving SSTs 1979–2008 from HadISST », as in the text. This means that the control is run with fixed climatological Jan and Feb LST, while the 10-member ensemble from which the climatological LST is computed is run with evolving LSTs and prescribed observed SSTs 1979–2008. Following the editor's comments we revise the text to clarify (**lines 147-148**).

-Line 175: what is the meaning of "lat" in the formula where describing the setting of cold integrations?
In **lines 154 and 300** we have corrected the smoothing function to
exp $\{- (\varphi- 38degN)^2 / (2(5deg)^2)\}$, where $\varphi$ latitude is greater than 38degN.

-Line 202: For the definition of EKE, it might be better to use the form $[(u^{HF})^2+ (v^{HF})^2]/2$.
We thank the editor for the suggestion, and have included the new form in **line 180**.

-Footnote 1: the definition of \theta_{cl} should be explained.
We have expanded the footnote to explain.

-Equations in line 201: if using z denoting the geopotential height, it should be $z^{HF}$ instead of $Z^{HF}$, and the meaning of $z^{HF}$ should be explained.
We have corrected to upper-case Z. The meaning of the HF superscript is defined in **lines 183-184**.

-The authors use 6 models' output (out of the 37 CMIP models) to make the "cold TP composite" which are actually from three modeling groups (CNRM, CanESM5 and FGOALS). It is possible that the "cold TP composite" still includes a considerable part of the inter-model difference. I suggest the authors at least add some discussion on the possible caveat of such composite.
We expand our comment on this caveat in **lines 124-129**, noting that a cold TP composite based on a selection of 1 model per institution shows consistent results.

REVIEWER #2

-L111: I assume these are the coupled simulations. Please make this clear.
This has been specified in **line 112**.

-L111: Why did you choose the years 1979-2008 for your analysis? Is this to have a 30-year period? Please add a sentence of clarity here.

The years constitute 30-year period representative of a recent climate. A brief explanation has been included in **line 113**.

-L190-193: I found this sentence very long and not worded that well. Please split into two sentences and rephrase to aid readability.
We thank the reviewer for pointing out the issue, and have reframed the sentence to aid readability. See **lines 199-203**.

-L191: 'especially'.
This has been corrected.

-Throughout degree symbols are missing when you refer to latitudes and longitudes.
We thank the reviewer for noting this, and have added the degree symbol where missing.

-L274/275: I believe you mean to reference figure 7 here.
We have corrected the mistake.